# TRAF2 regulates TNF and NF-κB signalling to suppress apoptosis and skin inflammation independently of Sphingosine kinase 1

Nima Etemadi[1,2,3,4], Michael Chopin[1,2], Holly Anderton[1,2], Maria C Tanzer[1,2], James A Rickard[1,2], Waruni Abeysekera[1], Cathrine Hall[1], Sukhdeep K Spall[1], Bing Wang[5], Yuquan Xiong[6], Timothy Hla[6], Stuart M Pitson[7], Claudine S Bonder[7], Wendy Wei-Lynn Wong[8], Matthias Ernst[3,4], Gordon K Smyth[1,9], David L Vaux[1,2], Stephen L Nutt[1,2], Ueli Nachbur[1,2], John Silke[1,2]*

[1]Walter and Eliza Hall Institute of Medical Research, Parkville, Australia; [2]Department of Medical Biology, University of Melbourne, Parkville, Australia; [3]Olivia Newton-John Cancer Research Institute, Heidelberg, Australia; [4]School of Cancer Medicine, La Trobe University, Heidelberg, Australia; [5]Center for Cardiovascular Research and Education in Therapeutics, Department of Epidemiology and Preventive Medicine, School of Public Health and Preventive Medicine, Monash University, Melbourne, Australia; [6]Center for Vascular Biology, Department of Pathology and Laboratory Medicine, Weill Cornell Medical College, Cornell University, New York, United States; [7]Centre for Cancer Biology, SA Pathology, Adelaide, Australia; [8]Institute of Experimental Immunology, University of Zurich, Zurich, Switzerland; [9]Department of Mathematics and Statistics, University of Melbourne, Parkville, Australia

*For correspondence: j.silke@latrobe.edu.au

Competing interests: The author declares that no competing interests exist.

**Abstract** TRAF2 is a component of TNF superfamily signalling complexes and plays an essential role in the regulation and homeostasis of immune cells. TRAF2 deficient mice die around birth, therefore its role in adult tissues is not well-explored. Furthermore, the role of the TRAF2 RING is controversial. It has been claimed that the atypical TRAF2 RING cannot function as a ubiquitin E3 ligase but counterclaimed that TRAF2 RING requires a co-factor, sphingosine-1-phosphate, that is generated by the enzyme sphingosine kinase 1, to function as an E3 ligase. Keratinocyte-specific deletion of *Traf2*, but not *Sphk1* deficiency, disrupted TNF mediated NF-κB and MAP kinase signalling and caused epidermal hyperplasia and psoriatic skin inflammation. This inflammation was driven by TNF, cell death, non-canonical NF-κB and the adaptive immune system, and might therefore represent a clinically relevant model of psoriasis. TRAF2 therefore has essential tissue specific functions that do not overlap with those of Sphk1.

## Introduction

TNF Receptor Associated Factor 2 (TRAF2) is an adaptor protein that transduces signals following ligation of certain cytokine receptors including those binding TNF. It was first identified together with TRAF1 as a component of TNF receptor-2 and then TNF receptor-1 (TNFR1) signalling complexes (*Rothe et al., 1994*; *Shu et al., 1996*). TRAF2, like most other TRAFs, contains a RING domain, several zinc fingers, a TRAF-N, and a conserved TRAF-C domain which is responsible for

**eLife digest** Psoriasis is an inflammatory disorder that causes red, flaky patches of skin. The disease affects around 2% of the world's population, and is most common in people of northern European descent. TNF is one of the key proteins in the development of psoriasis and drugs that inhibit TNF have been very successful in the treatment of this disease. However, these drugs are expensive and for unknown reasons at least 10% of patients do not respond to them.

Attempts to develop better drugs for psoriasis would be assisted by an improved understanding of this disease in terms of the genes and proteins involved. Etemadi *et al.* set out to obtain a more detailed molecular understanding of this disease by developing new mouse models of the condition. Mice were genetically engineered such that a key gene was deleted specifically from the skin cells that form the main barrier to the environment. These mice demonstrated that defects in skin cells called keratinocytes, rather than defects in the immune response, could lead to a psoriasis-like disease.

Etemadi *et al.* also showed that the skin cells with this genetic defect die in the presence of TNF and this cell death in mice caused a rapidly-appearing form of psoriasis. However, in the absence of TNF the mice still developed psoriasis, albeit more slowly. In this case, the condition was due to an excessive activation of a protein called NF-κB, which is known to play a role in maintaining balance in the immune system and in psoriasis.

These findings reveal how keratinocytes, cell death and inflammation can directly contribute to psoriasis-like conditions in mice. The next challenge will be to determine whether these findings can be used to help patients with this condition.

oligomerisation and receptor binding through its MATH region (*Takeuchi et al., 1996*; *Uren and Vaux, 1996*).

RING domains are nearly always associated with ubiquitin E3 ligase activity (*Shi and Kehrl, 2003*) and TRAF2 can promote ubiquitylation of RIPK1 in TNFR1 signalling complexes (TNFR1-SC) (*Wertz et al., 2004*). However TRAF2 recruits E3 ligases such as cIAPs to TNFR1-SC and these have also been shown to be able to ubiquitylate RIPK1 and regulate TNF signalling (*Dynek et al., 2010*; *Mahoney et al., 2008*; *Varfolomeev et al., 2008*; *Vince et al., 2009*). This makes it difficult to unambiguously determine the role of the E3 ligase activity of TRAF2.

Activation of JNK and NF-κB by TNF is reduced in cells from *Traf2*⁻/⁻ mice while only JNK signalling was affected in lymphocytes from transgenic mice that express a dominant negative (DN) form of TRAF2 that lacks the RING domain (*Lee et al., 1997*; *Yeh et al., 1997*). *Traf2*⁻/⁻*Traf5*⁻/⁻ mouse embryonic fibroblasts (MEFs) have a pronounced defect in activation of NF-κB by TNF, suggesting that absence of TRAF2 can be compensated by TRAF5 (*Tada et al., 2001*). Although activation of NF-κB was restored in *Traf2*⁻/⁻*Traf5*⁻/⁻ cells by re-expression of wild type TRAF2, it was not restored when the cells were reconstituted with TRAF2 point mutants that could not bind cIAPs (*Vince et al., 2009*; *Zhang et al., 2010*). These data, together with a wealth of different lines of evidence showing that cIAPs are critical E3 ligases required for TNF-induced canonical NF-κB (*Blackwell et al., 2013*; *Haas et al., 2009*; *Silke, 2011*), support the idea that the main function of TRAF2 in TNF-induced NF-κB is to recruit cIAPs to the TNFR1-SC. However, it remains possible that the RING of TRAF2 plays another function, such as in activating JNK and protecting cells from TNF-induced cell death (*Vince et al., 2009*; *Zhang et al., 2010*). Furthermore it has been shown that TRAF2 can K48-ubiquitylate caspase-8 to set the threshold for TRAIL or Fas induced cell death (*Gonzalvez et al., 2012*). Moreover, TRAF2 inhibits non-canonical NF-κB signalling (*Grech et al., 2004*; *Zarnegar et al., 2008*) and this function requires the RING domain of TRAF2 to induce proteosomal degradation of NIK (*Vince et al., 2009*). However, structural and in vitro analyses indicate that, unlike TRAF6, the RING domain of TRAF2 is unable to bind E2 conjugating enzymes (*Yin et al., 2009*), and is therefore unlikely to have intrinsic E3 ligase activity.

Sphingosine-1-phosphate (S1P) is a pleiotropic sphingolipid mediator that regulates proliferation, differentiation, cell trafficking and vascular development (*Pitson, 2011*). S1P is generated by sphingosine kinase 1 and 2 (SPHK1 and SPHK2) (*Kohama et al., 1998*; *Liu et al., 2000*). Extracellular S1P

mainly acts by binding to its five G protein-coupled receptors S1P$_{1-5}$ (*Hla and Dannenberg, 2012*). However, some intracellular roles have been suggested for S1P, including the blocking of the histone deacetylases, HDAC1/2 (*Hait et al., 2009*) and the induction of apoptosis through interaction with BAK and BAX (*Chipuk et al., 2012*).

Recently, it was suggested that the RING domain of TRAF2 requires S1P as a co-factor for its E3 ligase activity (*Alvarez et al., 2010*). Alvarez and colleagues proposed that SPHK1 but not SPHK2 is activated by TNF and phosphorylates sphingosine to S1P which in turn binds to the RING domain of TRAF2 and serves as an essential co-factor that was missing in the experiments of Yin *et al.* Alvarez and colleagues, observed that in the absence of SPHK1, TNF-induced NF-κB activation was completely abolished.

Although we know a lot about TRAF2, there are still important gaps particularly with regard to cell type specificity and in vivo function of TRAF2. Moreover, despite the claims that SPHK1 and its product, S1P, are required for TRAF2 to function as a ubiquitin ligase, the responses of *Traf2$^{-/-}$* and *Sphk1$^{-/-}$* cells to TNF were not compared. Therefore, we undertook an analysis of TRAF2 and SPHK1 function in TNF signalling in a number of different tissues.

Surprisingly, we found that neither TRAF2 nor SPHK1 are required for TNF mediated canonical NF-κB and MAPK signalling in macrophages. However, MEFs, murine dermal fibroblasts (MDFs) and keratinocytes required TRAF2 but not SPHK1 for full strength TNF signalling. In these cell types, absence of TRAF2 caused a delay in TNF-induced activation of NF-κB and MAPK, and sensitivity to killing by TNF was increased. Absence of TRAF2 in keratinocytes in vivo resulted in psoriasis-like epidermal hyperplasia and skin inflammation. Unlike TNF-dependent genetic inflammatory skin conditions, such as IKK2 epidermal knock-out (*Pasparakis et al., 2002*) and the *cpdm* mutant (*Gerlach et al., 2011*), the onset of inflammation was only delayed, and not prevented by deletion of TNF. This early TNF-dependent inflammation is caused by excessive apoptotic but not necroptotic cell death and could be prevented by deletion of *Casp8*. We observed constitutive activation of NIK and non-canonical NF-κB in *Traf2$^{-/-}$* keratinocytes which caused production of inflammatory cytokines and chemokines. We were able to reverse this inflammatory phenotype by simultaneously deleting both *Tnf* and *Nfkb2* genes. Our results highlight the important role TRAF2 plays to protect keratinocytes from cell death and to down-regulate inflammatory responses and support the idea that intrinsic defects in keratinocytes can initiate psoriasis-like skin inflammation.

## Results

### TRAF2 and SPHK1 are dispensable for TNF signalling in macrophages

Macrophages are a major source of TNF during inflammation and also respond to it by producing other inflammatory cytokines. To examine the role of TRAF2 in TNF signalling in macrophages we generated bone marrow-derived macrophages (BMDMs), from wild type and *Traf2$^{lox/lox}$Lyz2-Cre* (*Traf2$^{LC}$*) mice. The *Lyz2-Cre* recombinase transgene is expressed in all cells of the myeloid lineage including macrophages (*Wong et al., 2014*). TNF stimulation of wild type BMDMs caused degradation of IκBα and phosphorylation of JNK and ERK within 15 min (*Figure 1A*). In *Traf2$^{LC}$* and *Sphk1* deficient BMDMs, the kinetics of IκBα degradation and phosphorylation of JNK and ERK induced by TNF were the same as in wild-type BMDMs (*Figure 1A and 1B*).

Because activation of canonical NF-κB protects cells from TNF-induced death, we treated *Traf2*-deficient and wild-type BMDMs with TNF for 24 hr and assayed their viability using Propidium Iodide (PI) staining and flow cytometry. Taking into account the slightly higher background death of *Traf2$^{-/-}$* BMDMs compared with wild type BMDMs, TNF was not cytotoxic to either wild-type or *Traf2$^{-/-}$* macrophages (*Figure 1C*). High-dose (500 nM) Smac-mimetic (SM) but not a low dose (100 nM) can induce the death of macrophages via production of autocrine TNF (*Wong et al., 2014*). We tested whether loss of TRAF2 could sensitise macrophages to low dose SM. As previously reported, wild-type macrophages were resistant to killing by low-dose SM, but *Traf2$^{LC}$* BMDMs were very sensitive (*Figure 1C*). To find out why, we measured the concentration of TNF secreted by BMDMs into the culture media following stimulation with TNF or SM. Increased TNF secretion was undetectable earlier than 20 hr after TNF stimulation and there was no significant difference in the amount of TNF secreted by wild type or *Traf2$^{LC}$* macrophages at this time point (*Figure 1D*; upper panel). In contrast, after 4 hr of treatment with SM, very large amounts of TNF could be detected in both the

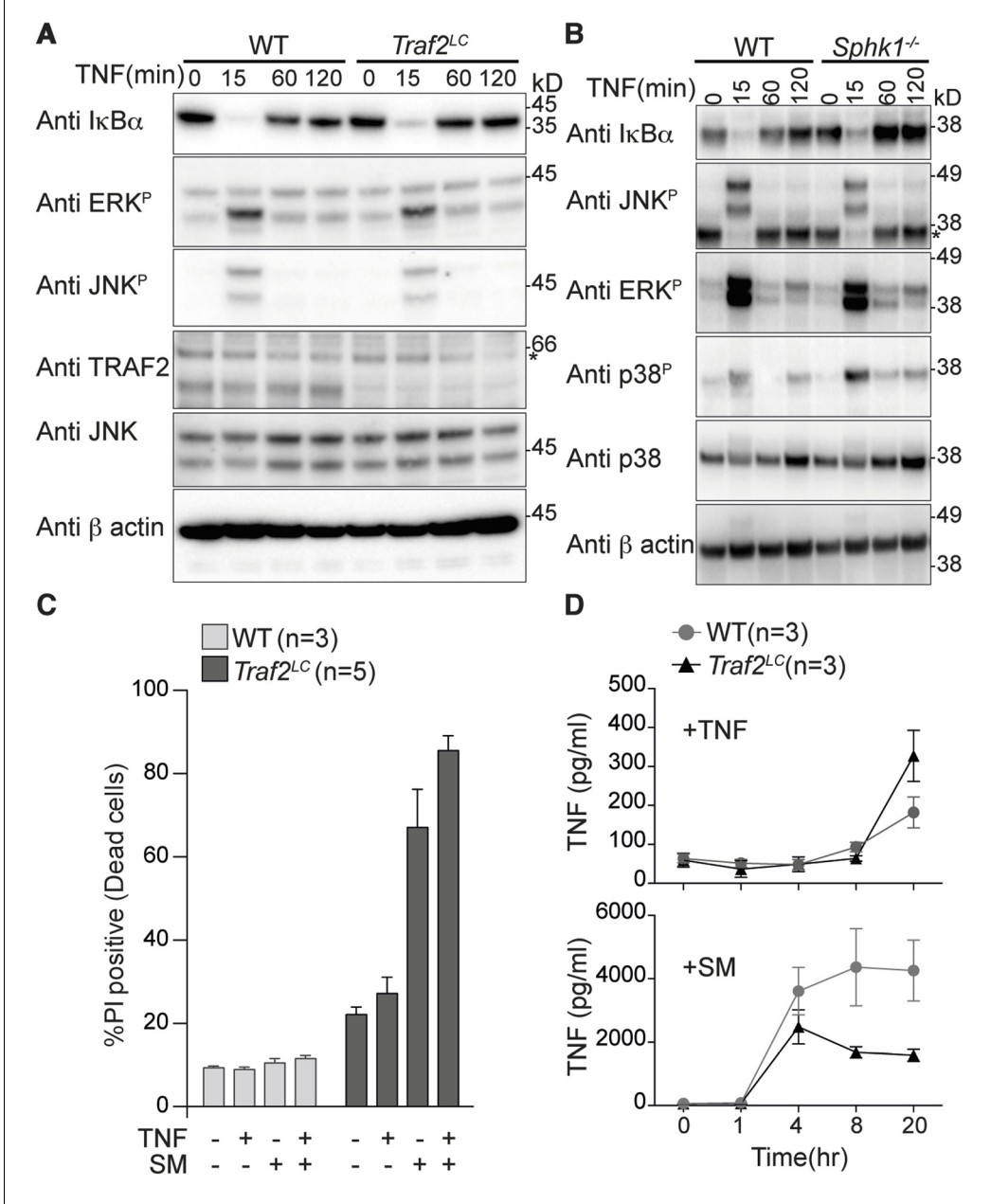

**Figure 1.** TRAF2 is not required for TNF-induced NF-κB and MAPK signalling in bone marrow derived macrophages. (**A** and **B**) Western blot analysis of wild type (WT), *Traf2^lox/lox Lyz2-Cre* (*Traf2^LC*) and *Sphk1^-/-* BMDMs treated with TNF (20 ng/ml) for the indicated times. (**C**) Flow cytometric analysis of WT, *Traf2^LC* BMDMs treated with TNF (20 ng/ml) ± 100 nM Smac-mimetic (SM) for 24 hr and stained with Propidium Iodide (PI) and analysed by flow cytometry. (**D**) WT, *Traf2^LC* macrophages were treated with TNF (20 ng/ml) or Smac-mimetic (100 nM) for indicated time and supernatants analysed by TNF ELISA. Data are represented as mean ± SEM.

wild-type and *Traf2^LC* macrophages (*Figure 1D*; lower panel). Although initially similar amounts of TNF were produced, after 8 hr the amount of TNF secreted by the *Traf2^LC* macrophages was less than that made by the wild-type macrophages (*Figure 1D*; lower panel). This is probably due to SM-induced death of *Traf2^LC* but not wild-type BMDMs (*Figure 1C*).

## TRAF2 but not SPHK1 is needed for normal TNF signalling in fibroblasts and keratinocytes

Since TRAF2 was not required for TNF signalling in BMDMs, we could not use this cell type to determine whether SPHK1-derived S1P was required as an essential co-factor for TRAF2s E3 ligase activity. Therefore, we extended our analysis to other cell types and generated MEFs and MDFs lacking TRAF2 or SPHK1 and compared their response to TNF with wild-type cells. Consistent with earlier reports (*Tada et al., 2001*; *Vince et al., 2009*; *Zhang et al., 2010*), loss of TRAF2 in MEFs caused a delay in, but did not abolish, TNF-induced NF-κB activation and JNK phosphorylation (*Figure 2A*). Similarly, loss of TRAF2 in MDFs caused a delay in both NF-κB and MAPK signalling activation (*Figure 2B*). However, no such delay was observed in *Sphk1* knock-out cells (*Figure 2A and 2B*). Consistent with earlier findings (*Vince et al., 2009*; *Zhang et al., 2010*), *Traf2* knock-out MEFs and MDFs were sensitive to TNF-induced cell death; however, wild-type and *Sphk1*$^{-/-}$ cells remained resistant (*Figure 2C*). The sensitivity to TNF-induced cell death correlated with caspase-8 and -3 cleavage in *Traf2*$^{-/-}$ fibroblasts (*Figure 2B*).

To test the function of TRAF2 in primary keratinocytes and compare its role to that of SPHK1 in TNF signalling, we generated epidermal specific *Traf2*-deficient mice (*Traf2*$^{EKO}$) by crossing *Traf2*$^{lox/lox}$ mice with transgenic mice constitutively expressing Cre recombinase under the control of the keratin-14 promoter (*K14-Cre*). Primary keratinocytes isolated from wild-type, *Traf2*$^{EKO}$ and *Sphk1*$^{-/-}$ mice were tested and, similarly to MEFs and MDFs, TNF signalling was delayed and weakened in *Traf2*-deficient (*Figure 2D*), but not *Sphk1*-deficient keratinocytes (*Figure 2E*). *Traf2*$^{-/-}$, but not *Sphk1*$^{-/-}$, keratinocytes were also sensitive to TNF-induced death (*Figure 2F*), and this was associated with a rapid reduction in cFLIP$_L$ levels and caspase-8 cleavage (*Figure 2D*). Treatment with the pan-caspase inhibitor Q-VD-OPh (QVD) reduced TNF-induced death of *Traf2*$^{-/-}$ cells by about half (*Figure 2F*). On its own, the RIPK1 kinase inhibitor, Necrostatin (Nec), reduced TNF-induced cell death, but this reduction is not statistically significant. However, the combination of QVD and Nec completely blocked TNF-induced cell death in *Traf2*$^{-/-}$ keratinocytes (*Figure 2F*).

## TNF induced RIPK1 ubiquitylation is delayed in *Traf2*$^{-/-}$ MDFs, but normal in *Sphk1*$^{-/-}$ cells

Alvarez et al. proposed that SPHK1 and its product S1P are required for ubiquitylation of RIPK1 by TRAF2 (*Alvarez et al., 2010*). However, the preponderance of evidence, both genetic and biochemical, suggests that TRAF2 serves to recruit cIAPs and that these are the main E3 ligases for RIPK1 (*Mahoney et al., 2008*; *Silke, 2011*; *Varfolomeev et al., 2008*; *Vince et al., 2009*). To explore this issue further, we examined RIPK1 ubiquitylation upon TNF stimulation in wild-type, *Traf2*$^{-/-}$ and *Sphk1*$^{-/-}$ MDFs using the tandem ubiquitin binding entities (TUBE) beads to precipitate ubiquitylated proteins. Within 5 to 15 min of TNF stimulation RIPK1 became strongly ubiquitylated in both wild-type and *Sphk1*$^{-/-}$ MDFs, and this decreased over 60 min (*Figure 3A and 3B*). In *Traf2*$^{-/-}$ cells, this ubiquitylation was not observed at early time points (5 or 15 min), but occurred after 60 min with a different pattern (*Figure 3A and 3B*). This delayed ubiquitylation correlated with the delayed NF-κB activation observed in *Traf2*$^{-/-}$ cells (*Figure 2A,B and 2D*). cIAP1 and phospho-ERK migrated as high molecular weight species, suggestive of ubiquitylation, within 15 min of TNF stimulation in wild type and *Sphk1*$^{-/-}$, but this ubiquitylation was delayed in *Traf2*$^{-/-}$ MDFs (*Figure 3A and 3B*). Consistent with the previous results, cleaved caspase-8 appeared in *Traf2*$^{-/-}$ MDFs after 1 hr TNF stimulation (*Figure 3B*).

## Loss of TRAF2 in keratinocytes causes epidermal hyperplasia and psoriasis-like skin inflammation

The role of TRAF2 in homeostasis of lymphocytes has been well-studied (*Silke and Brink, 2010*). However, the role of TRAF2 in adult epithelial cells has not yet been reported. TRAF2 was detected at high or medium levels in 36 out of 81 normal human tissue cell types, and in the skin, dermal fibroblasts and keratinocytes (*Uhlen et al., 2015*). To discover the role of TRAF2 in keratinocytes we aged and monitored *Traf2*$^{lox/lox}$-*K14-Cre* (*Traf2*$^{EKO}$) mice. *Traf2*$^{EKO}$ mice were born at the expected Mendelian ratios but developed an inflamed skin phenotype from 6 weeks of age. This phenotype became worse over time and the mice were sacrificed usually before 15 weeks of age (*Figure 4A and 4F*). The epidermal layer in a normal mouse is usually composed of 2–3 layers of keratinocytes

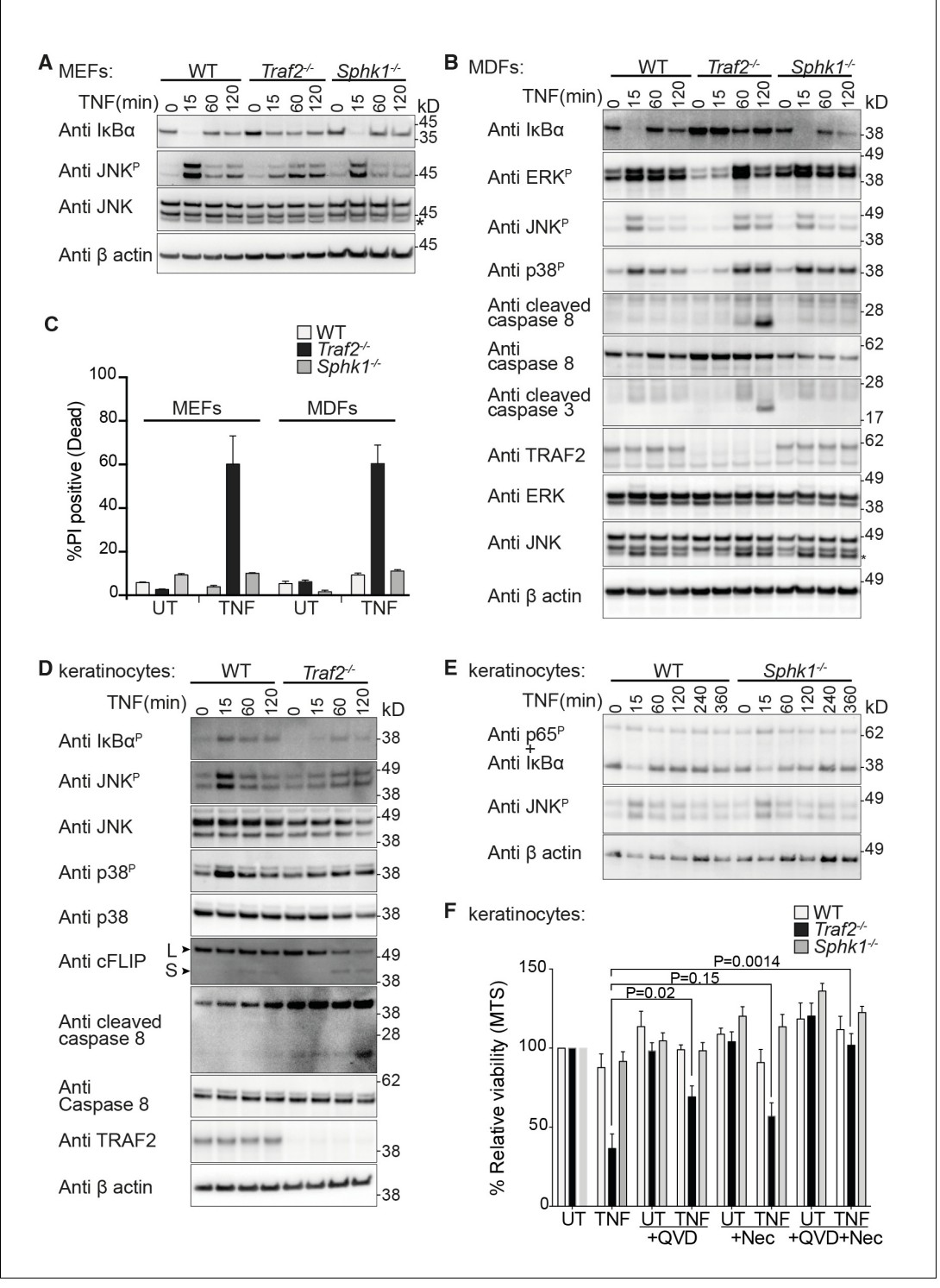

**Figure 2.** TRAF2 but not SPHK1 is required for TNF signalling in MEFs, MDFs and keratinocytes. (**A** and **B**) Western blot analysis of wild-type (WT), *Traf2*−/− and *Sphk1*−/− immortalised MEFs and MDFs (as indicated) treated with TNF (100 ng/ml) for the indicated times. (**C**) FACS analysis of WT, *Traf2*−/− and *Sphk1*−/− MEFs and MDFs treated with TNF (100 ng/ml) for 24 hr and stained with Propidium Iodide (PI). n≥3 biological repeats. Data are represented as mean ± SEM. (**D**) Western blot analysis of WT and *Traf2*−/− keratinocytes treated with TNF (100 ng/ml) for the indicated times. (**E**) Western blot analysis of WT and *Sphk1*−/− keratinocytes treated with TNF (100 ng/ml) for the indicated times. (**F**) WT, *Traf2*−/− and *Sphk1*−/− keratinocytes were treated with TNF (100 ng/ml) in the absence or presence of Q-VD-OPh (QVD; 10 µM) and Necrostatin (Nec; 50 µM) as indicated for 24 hr and

*Figure 2 continued on next page*

*Figure 2 continued*

metabolically active cells were measured by MTS-PMS (MTS) assay. Biological repeats were WT and *Traf2*[-/-] (n=5) and *Sphk1*[-/-] (n=3). Data are represented as mean ± SEM. P values compare the indicated samples using a Students t-test.

(*Gudjonsson et al., 2007*); however, histological skin sections from *Traf2*[EKO] mice revealed a grossly thickened epidermis (*Figure 4B*). Normally, the epidermis expresses keratin-14, but keratin-6 expression is confined to hair follicles or hyper-proliferative keratinocytes. The *Traf2*[EKO] epidermis was highlighted by widespread expression of both of these markers (*Figure 4B*). Excessive leukocyte infiltration and apoptotic death were also detected in the inflamed area of *Traf2*[EKO] by immunostaining of CD45 (leukocyte common antigen) and cleaved caspase-3 (CC3), respectively (*Figure 4B*). We did not detect any significant infiltration of mast cells into the inflamed area of *Traf2*[EKO] mice by toluidine blue staining (*Figure 4B*).

## Skin inflammation induced by keratinocyte specific loss of TRAF2 partially depends on TNF

Previous studies have shown that defective TNF signalling leads to inflammation in the skin, and deletion of TNF or TNFR1 prevents this phenotype. In the *cpdm* (SHARPIN mutant) mouse, loss of one allele of *Tnf* significantly reduced the systemic inflammatory disorder (*Gerlach et al., 2011*). Based on the presence of apoptotic cells in the epidermal layer of *Traf2*[EKO] mice (*Figure 4B*), spontaneous secretion of TNF by *Traf2*[-/-] keratinocytes in culture (*Figure 4C*) and the disrupted TNF signalling in *Traf2*[-/-] keratinocytes (*Figure 2D*), we hypothesised that depletion of TNF would also prevent the inflammation in the skin of *Traf2*[EKO] mice. Therefore, *Traf2*[EKO] mice were crossed to *Tnf*[-/-] mice. Consistent with our hypothesis, *Traf2*[EKO]*Tnf*[-/-] mice did not develop epidermal hyperplasia and skin inflammation at the same time as *Traf2*[EKO] mice (*Figure 4D*). However, at ~20 weeks after birth (10 weeks later than *Traf2*[EKO]), the skin of most of the *Traf2*[EKO]*Tnf*[-/-] mice started to become inflamed. Lesions became severe by 30 weeks, and these necessitated sacrifice of the affected animals (*Figure 4D and 4F*).

Histological analysis revealed that leukocyte infiltration into the dermis of 10-week-old *Traf2*[EKO]*Tnf*[-/-] mice was also reduced when compared to age-matched *Traf2*[EKO] mice, but increased by 23 weeks (*Figure 4E*). There was no evidence for apoptotic, cleaved caspase 3 (CC3)-positive cells in the skin of *Traf2*[EKO]*Tnf*[-/-] mice at any age, suggesting that cell death is induced by TNF in the *Traf2*[-/-] mice (*Figure 4B*) and that increased proliferation can be genetically separated from apoptotic cell death.

## Apoptotic cell death induces early onset skin inflammation in *Traf2*[EKO] mice

Because TNF-induced death in *Traf2*[-/-] keratinocytes was only partially attenuated by Q-VD-OPh but was completely blocked by the combination of Q-VD-OPh and Necrostatin (*Figure 2F*), we hypothesised that keratinocytes might be undergoing a necroptotic cell death that could contribute to skin inflammation (*Pasparakis and Vandenabeele, 2015*). However, depletion of MLKL, an essential necroptosis effector (*Murphy et al., 2013*; *Sun et al., 2012*), did not prevent or alter the onset of the skin inflammation in *Traf2*[EKO] mice (*Figure 5A and 5B*). Deficiency of both MLKL and caspase-8 did, however, rescue the early skin inflammation in *Traf2*[EKO] (*Figure 5A and 5B*) but the mice could not be aged further because *Mlkl*[-/-]*Casp8*[-/-] mice develop lymphadenopathy before 15 weeks of age (Alvarez-Diaz *et al.*, personal communication, October 2015). To molecularly define the type of TNF-induced cell death in *Traf2*[-/-] keratinocytes, we looked at the necroptotic and apoptotic markers upon TNF stimulation in detail (*Figure 5C and 5D*). Oligomerisation of MLKL in the membrane fraction is a marker for execution of necroptosis (*Hildebrand et al., 2014*) and did not occur following 5 hr of TNF stimulation, but was only seen when caspases were inhibited by Q-VD-OPh (*Figure 5C*). As expected, however, oligomerisation of MLKL induced by TNF and Q-VD-OPh treatment was blocked by Necrostatin (*Figure 5C*).

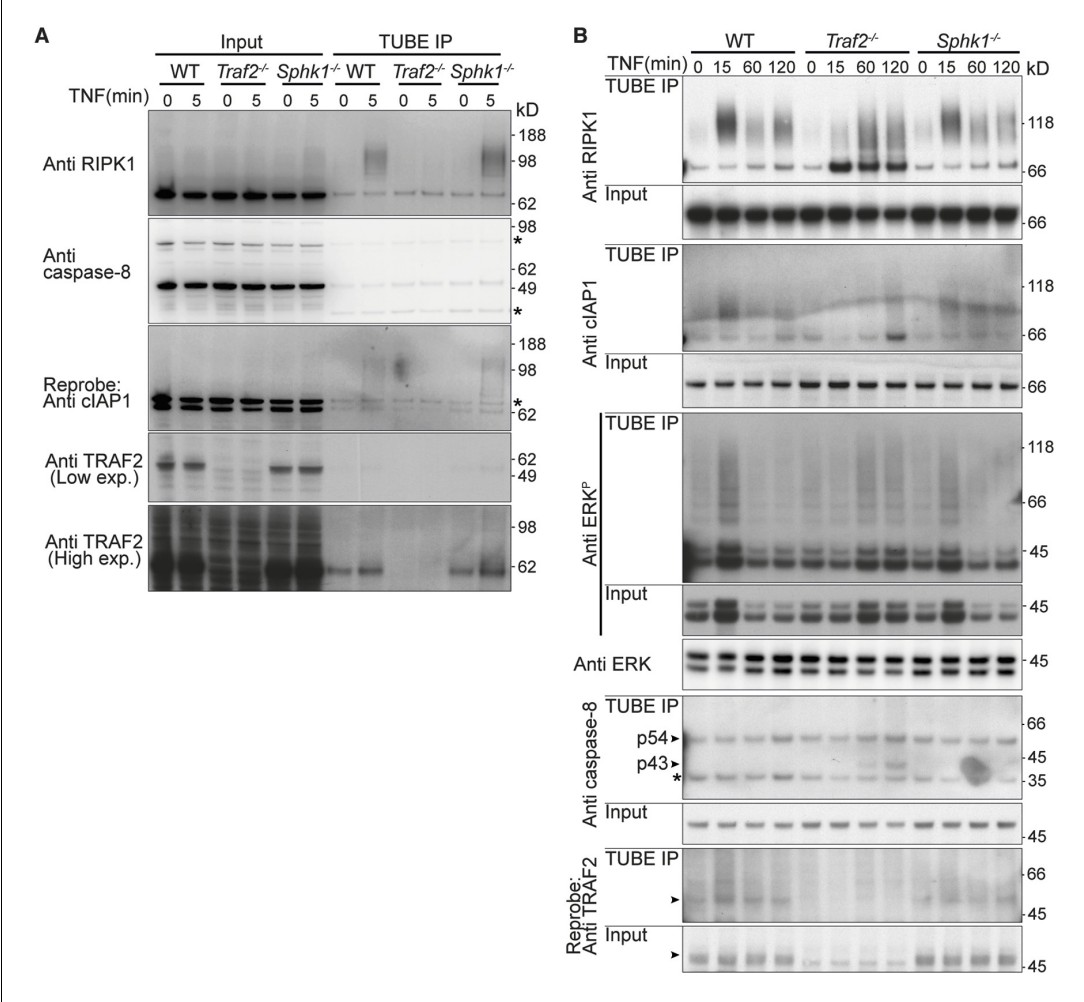

**Figure 3.** TRAF2 but not SPHK1 is required for rapid, TNF-induced, RIPK1 ubiquitylation. (**A** and **B**) Wild-type, *Traf2⁻/⁻* and *Sphk1⁻/⁻* immortalised MDFs were treated with TNF (100 ng/ml) for the indicated times. Cell lysates were prepared and precipitated with ubiquitin-binding TUBE beads, then separated on an SDS-PAGE gel and Western blotted with the indicated antibodies.

## Psoriasis-like inflammation in *Traf2^EKO* mice is characterised by skin infiltration of neutrophils and IFNγ producing CD4⁺ T cells

To characterise the type of inflammation in *Traf2^EKO* and *Traf2^EKO Tnf⁻/⁻* mice, we analysed skin sections of 10-week-old *Traf2^EKO* and 20-week-old *Traf2^EKO Tnf⁻/⁻* for infiltrating inflammatory leukocytes, using flow cytometry. The skin from *Traf2^EKO* and *Traf2^EKO Tnf⁻/⁻* mice had higher numbers of CD11b⁺Ly6G⁺ neutrophils in the epidermal and dermal layers compared to wild-type skin (*Figure 6A,B,C and 6D*). CD4⁺ T cells were also abundant in the inflamed skin of *Traf2^EKO* and *Traf2^EKO Tnf⁻/⁻* (*Figure 6E and 6F*). Surprisingly, IFNγ producing CD4⁺ T cells were significantly more abundant in the skin of *Traf2^EKO Tnf⁻/⁻* than *Traf2^EKO* (*Figure 6G*). To investigate whether IFNγ production by these T cells contributed to the inflammatory phenotype, we lethally irradiated *Traf2^EKO* mice and reconstituted them with bone marrow cells from *Traf2^EKO* (which harbour wild-type bone marrow cells) or *Ifng⁻/⁻* mice. However, both wild type and *Ifng⁻/⁻* chimeras developed skin inflammation to the same extent, indicating that hematopoietic cell-derived IFNγ is not a major contributor to the psoriasis-like skin inflammation (*Figure 7*).

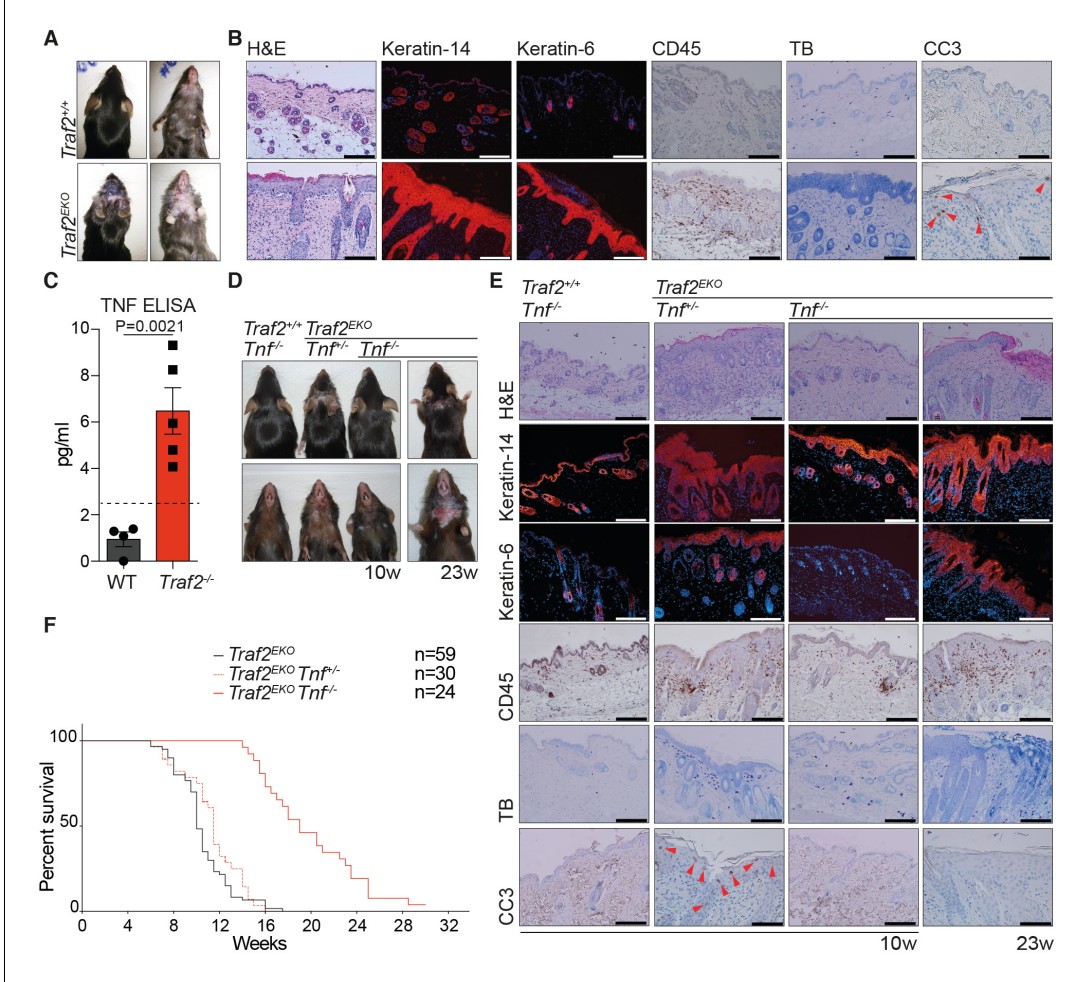

**Figure 4.** Loss of TRAF2 in keratinocytes causes epidermal hyperplasia and psoriasis-like skin inflammation. (**A**, **B**, **D** and **E**) Representative images and skin sections of the indicated mice strains stained with haematoxylin/eosin (H&E), immuno-histochemistry for the pan leukocyte marker (CD45), cleaved caspase-3 (CC3), or toluidine blue (TB), or immuno-stained for Keratin-14 or Keratin-6 (in red) plus Hoechst (nuclei in blue). Scale bars = 100 μm. (**C**) The culture media of primary keratinocytes of the indicated genotypes at 80% confluency was replaced with serum-free media which was collected 48 hr later and analysed with TNF ELISA. Biological repeats WT (n=4) and *Traf2*⁻/⁻ (n=5) are indicated. Data are represented as mean ± SEM. The dashed line indicates minimum detectable range of the ELISA. P values compare the indicated samples using a Students t-test. (**F**) Kaplan-Meier graph depicting survival of indicated mouse strains.

## Loss of TRAF2 causes constitutive non-canonical NF-κB activation and expression of inflammatory cytokines in keratinocytes

As TNF depletion only delayed and did not prevent the inflammatory skin phenotype and there were infiltrating leukocytes in the skin of *Traf2*ᴱᴷᴼ*Tnf*⁻/⁻ mice, this implied that there were factor(s) expressed by keratinocytes that attracted these inflammatory cells independently of TNF. TRAF2 suppresses non-canonical NF-κB activation in many different cell types and loss of TRAF2 induces constitutive activation of non-canonical NF-κB (*Grech et al., 2004*; *Lin et al., 2011*; *Vallabhapurapu et al., 2008*; *Vince et al., 2007*; *Zarnegar et al., 2008*). We therefore hypothesised that loss of *Traf2* in keratinocytes might induce activation of non-canonical NF-κB and thereby drive expression of inflammatory genes. To test this hypothesis we analysed the status of NIK and NFκB2/ p100 processing in *Traf2*⁻/⁻ keratinocytes. Consistent with non-canonical NF-κB activation, NIK was readily detectable in *Traf2*⁻/⁻ but not wild-type keratinocytes and there were higher levels of p52 in these cells. This activation of non-canonical NF-κB was likely independent of TNF because stimulation of these cells for 2 hr with TNF did not change the level of NIK or p52 (*Figure 8A*). To investigate whether increased non-canonical NF-κB caused an increase in keratinocyte proliferation, which

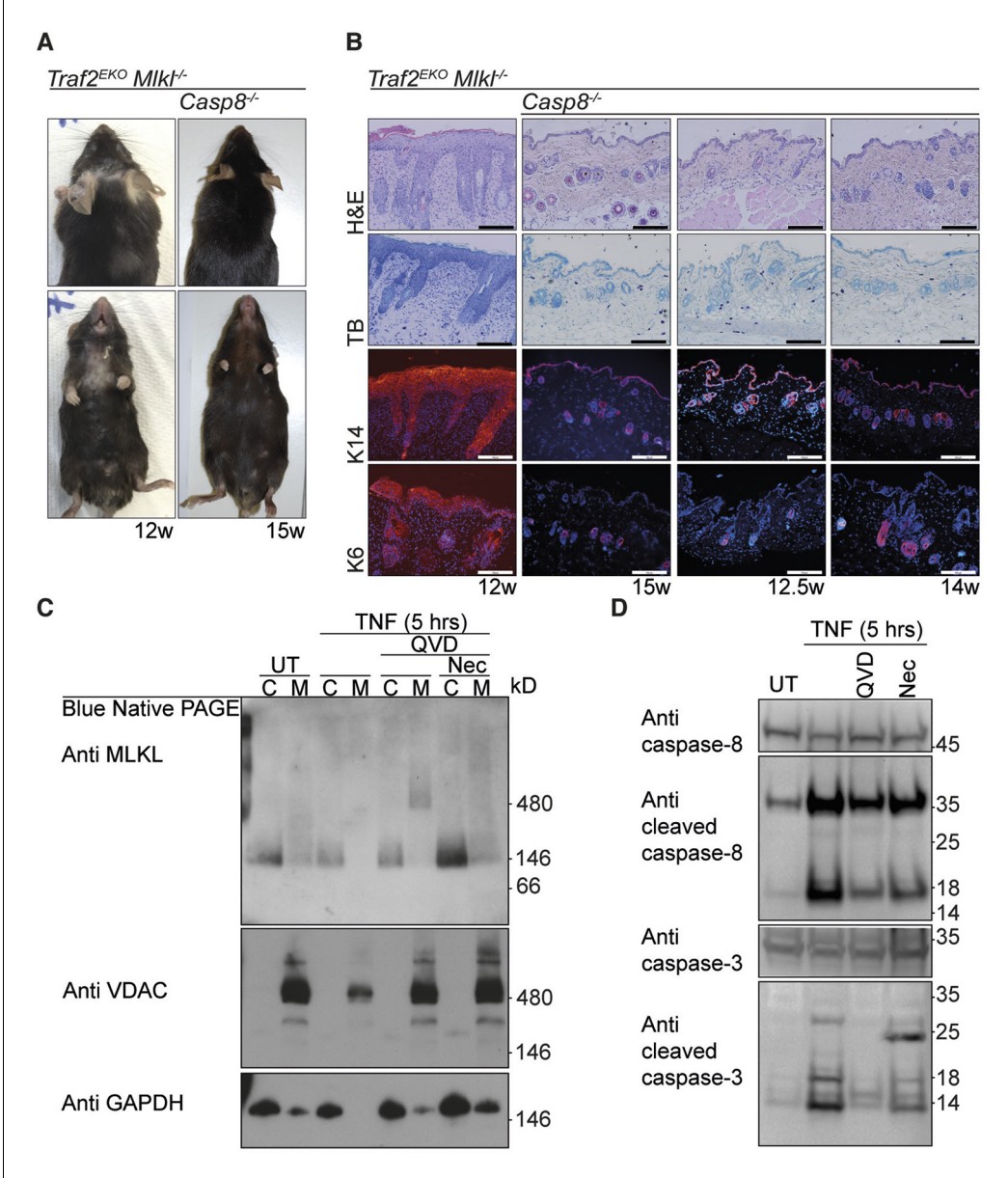

**Figure 5.** The early inflammation in *Traf2^EKO* is caused by keratinocyte apoptosis. (A and B) Representative images and skin sections of indicated mice stained with haematoxylin/eosin (H&E), toluidine blue (TB) or immuno-stained for Keratin-14 or Keratin-6 (in red) plus Hoechst (nuclei in blue). Scale bars = 100 μm. (C) *Traf2^-/-* and *Sphk1^-/-* keratinocytes were treated with TNF (100 ng/ml) for 5 hr in the absence or presence of QVD and/or Nec as indicated. Cytoplasmic (C) and Membrane (M) fractions were separated on a Blue Native PAGE gel and the Western blot probed with the indicated antibodies. (D) *Traf2^-/-* keratinocytes were treated with TNF (100 ng/ml) for 5 hr in the absence or presence of QVD or Nec as indicated. The lysates were separated on an SDS-PAGE gel and the Western blot probed with the indicated antibodies.

might account for the thickened epidermis, we measured the numbers of metabolically active cells every day over 5 days using an MTS-PMS assay in wild-type and *Traf2^-/-* keratinocytes on a *Tnf^-/-* background (as a surrogate marker for proliferation). However, no difference was observed in the proliferation of *Traf2^-/-* versus wild-type keratinocytes (*Figure 8B*). To simultaneously control for the genotypes of the keratinocytes and the accuracy of the MTS assay, keratinocytes were treated with TNF for 24 hr after 4 days in culture and the viability of the cells were measured on day 5 with MTS assay. As expected, *Traf2^-/-* keratinocytes were sensitive to TNF and gave a very low MTS value,

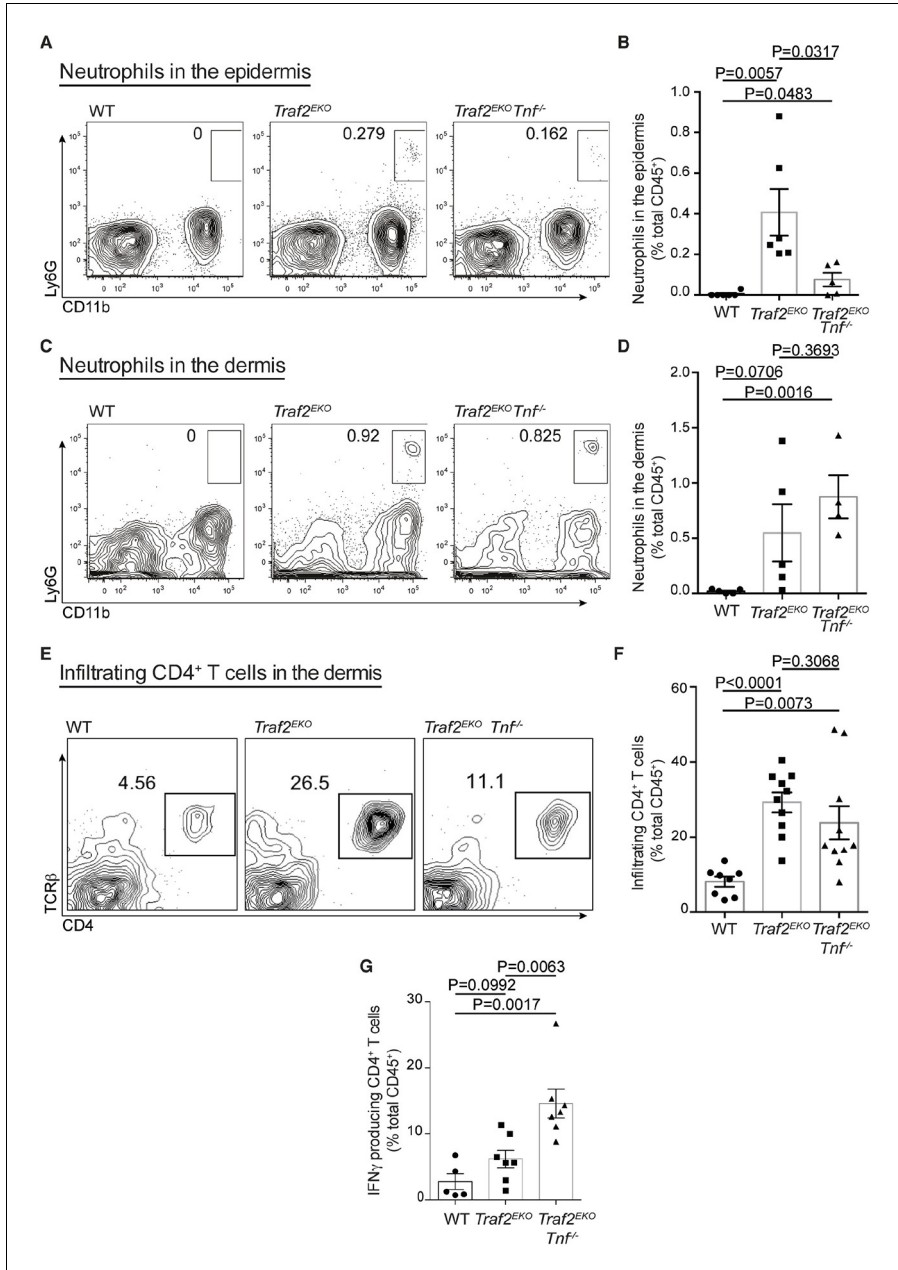

**Figure 6.** Loss of TRAF2 in keratinocytes causes infiltration of neutrophils and IFNγ-producing CD4+ T cells to the skin. (A-G) Single-cell suspensions from mouse ears with skin inflammation and healthy controls were prepared and the haematopoietic component (CD45.2+ and PI-) analysed by flow cytometry using the indicated markers. Representative contour plots for Ly6G and CD11b staining in the epidermis (A) and dermis (C) of mice with indicated genotype. Numbers indicate the proportion of CD11b+Ly6G+ neutrophils (boxed). Bar graph showing percentage infiltrated neutrophils to the epidermis (B) and dermis (D) n≥5, error bars are SEM. (C) Representative contour plots for TCRβ and CD4 staining. Numbers indicate the proportion of CD4+ T cells (boxed). (E) Representative contour plots for CD4 and TCRβ staining in the dermis of mice with indicated genotype. (F) Bar graph showing the percent of skin infiltrating CD4+ T cells, n≥8, error bars are SEM. (G) Bar graph showing percentage of infiltrating IFNγ-producing CD4+ T cells to the dermis of mice indicated genotype, n≥5, data are represented as mean ± SEM. Symbols in the graphs represent individual mice. All data are relative to the total CD45+ and PI- cells. P values compare the indicated samples using a Students t-test.

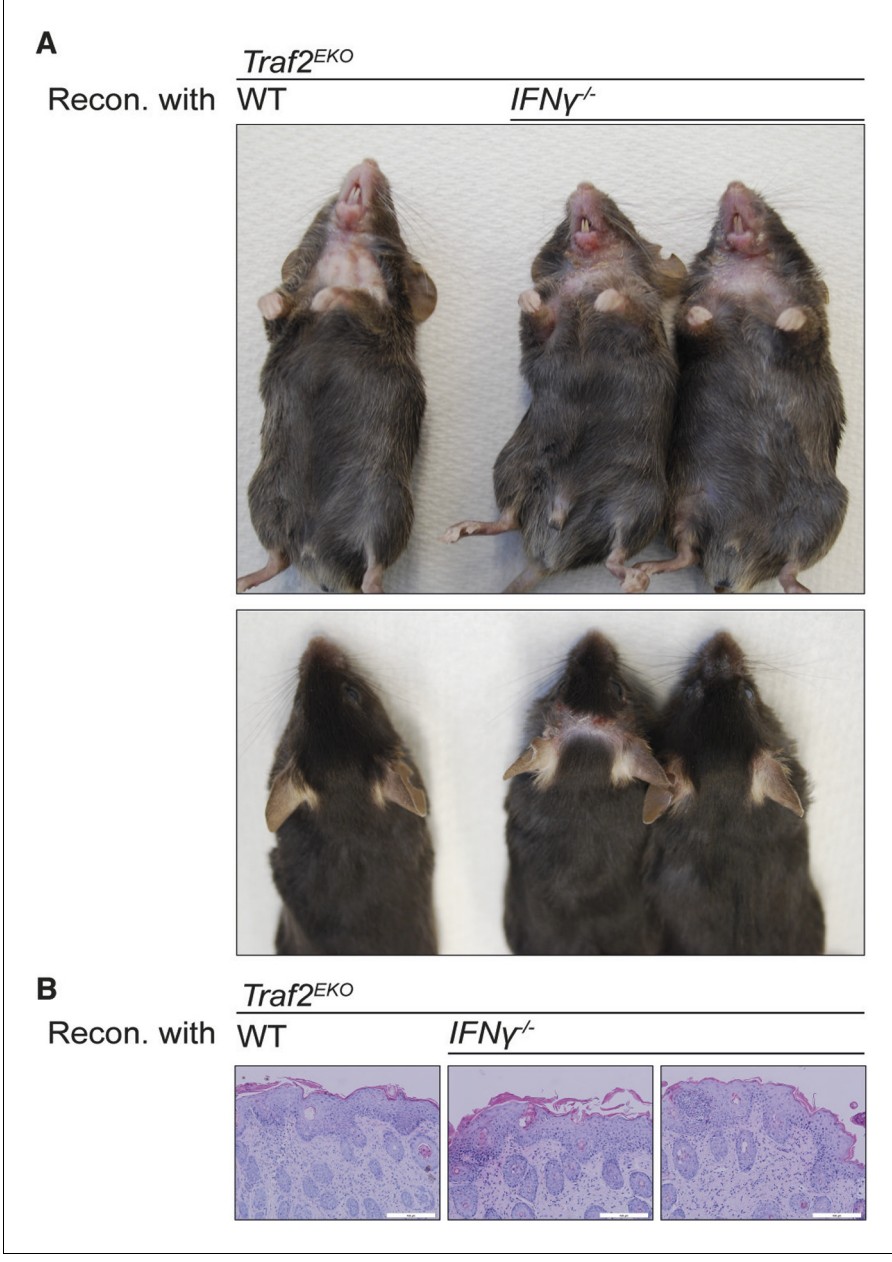

**Figure 7.** Reconstitution with IFNγ-deficient bone marrow does not prevent the inflammation in *Traf2^EKO* mice. (**A,B**) 6-week-old *Traf2^EKO* mice were lethally irradiated and injected with wild-type (wt) or *Ifnγ^-/-* bone marrow cells. After 4 weeks of reconstitution (Recon.), all mice, regardless of the origin of injected cells, started to develop inflammation similarly and needed to be sacrificed by 8 weeks after reconstitution. (**A**) Representative images. (**B**) H&E staining of skin sections. Scale bars = 100 µm.

while *Traf2^+/-* keratinocytes, generated from littermate controls, were resistant to TNF-induced death (*Figure 8B*).

IL-17 plays a major role in psoriasis and has been shown to induce proliferation of keratinocytes (*Ha et al., 2014*). On the other hand TRAF3 can inhibit IL-17 signalling (*Zhu et al., 2010*). Considering that TRAF2 and 3 can act together (*Gardam et al., 2008*; *Jabara et al., 2002*; *Vallabhapurapu et al., 2008*; *Zarnegar et al., 2008*), we investigated whether loss of TRAF2 affected IL-17 signalling. However, loss of TRAF2 did not affect IL-17 signalling in *Traf2^-/-* keratinocytes, as assessed by IκBα degradation and phosphorylation of p65 and p38

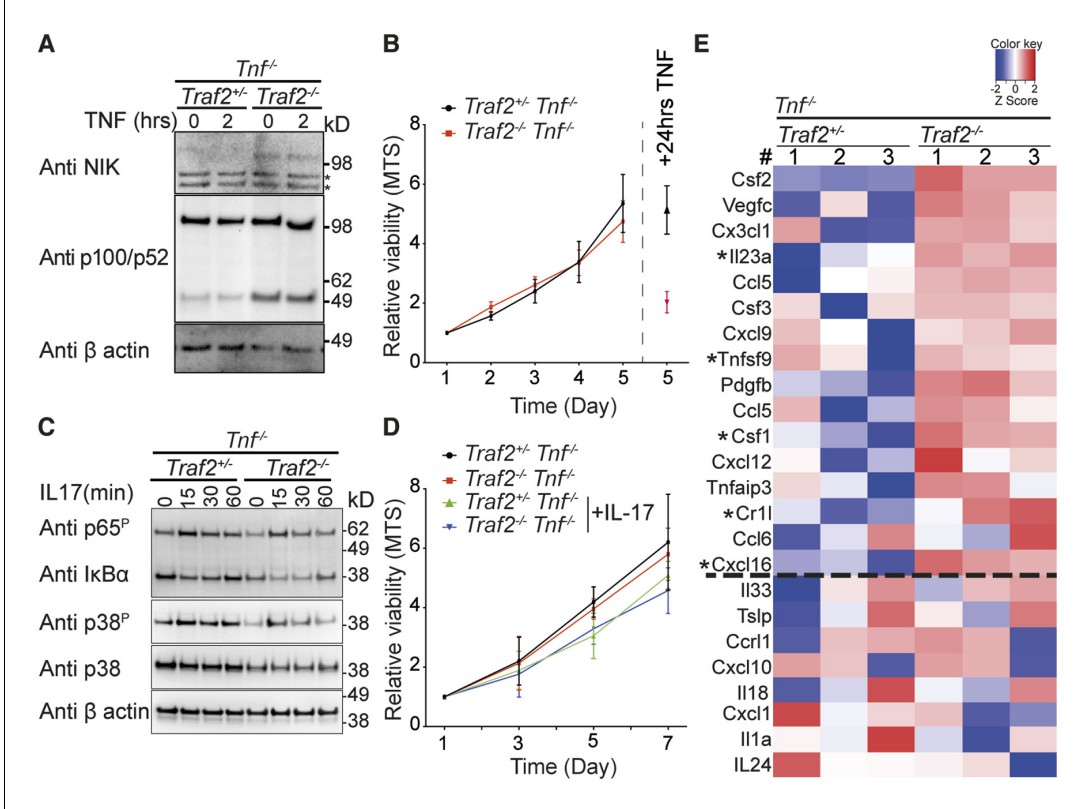

**Figure 8.** TRAF2 deletion causes constitutive activation of the non-canonical NF-κB transcription factor pathway. (A) $Tnf^{-/-}Traf2^{+/-}$ and $Tnf^{-/-}Traf2^{-/-}$ keratinocytes were treated with or without TNF (100ng/ml) for the indicated times. The lysates were analysed as in **Figure 1**. Asterisks indicate non-specific bands. (B) The viability of keratinocytes of the indicated genotype was measured at indicated time points using an MTS assay. Viability following TNF treatment (100 ng/ml) on day 4 was used as a control. Data are represented as mean ± SEM, n=4 biological repeats. (C) $Tnf^{-/-}Traf2^{+/-}$ and $Tnf^{-/-}Traf2^{-/-}$ keratinocytes were treated with IL-17 (100 ng/ml) for the indicated times and lysates analysed as previously. (D) Viability of keratinocytes of indicated genotype ± IL-17 (100 ng/ml) was measured at indicated time points using MTS assay. Data are represented as mean ± SEM, n≥3 biological repeats. (E) Heat map depicting a selection from qPCR analysis of more than 600 inflammatory genes from $Tnf^{-/-}Traf2^{+/-}$ and $Tnf^{-/-}Traf2^{-/-}$ keratinocytes (3 mice for each genotype). Log expression values have been standardized to have mean 0 and standard deviation 1 for each row. Genes were ranked based on the fold change expression. Gens above the dashed line were highly elevated in $Tnf^{-/-}Traf2^{-/-}$ keratinocytes. Asterisks indicate significant changes with P values <0.05.

The following source data is available for figure 8:

**Source data 1.** Complete zoom-able heat map of qPCR array.

**Source data 2.** Spreadsheet of result from qPCR array.

(**Figure 8C**). Moreover, stimulation of keratinocytes by IL-17 did not induce proliferation of keratinocytes either in wild-type or $Traf2^{-/-}$ keratinocytes in vitro (**Figure 8D**).

To examine the impact of constitutive activation of non-canonical NF-κB transcription factor on the expression of inflammatory genes in keratinocytes, we used a qPCR 'inflammation array' to measure more than 600 inflammatory genes. We isolated and cultured keratinocytes from $Traf2^{EKO/+}$ and $Traf2^{EKO}$ mice both on a $Tnf^{-/-}$ background to avoid the effect of autocrine TNF. After 48 hr starvation (to exclude any extrinsic signal by the serum) and without any stimulation (to look at the intrinsic effect of non-canonical NF-κB transcription factor), cells were lysed and their RNA were extracted and analysed by the qPCR array. Out of more than 600 inflammatory genes analysed, about 300 genes were constitutively expressed in the keratinocytes under these conditions (**Figure 8E**). The signature of inflammatory genes expression was changed in $Traf2^{-/-}Tnf^{-/-}$ compared to $Traf2^{+/-}Tnf^{-/-}$ keratinocytes (**Figure 8E**). Many inflammatory genes, such as Csf1 (M-CSF), IL-23,

TNFSF9 (4-1BBL), Cr1l and Cxcl16, were highly elevated in $Traf2^{-/-}Tnf^{-/-}$ keratinocytes (an Excel list and zoom-able heat map of 338 expressed genes are available as *Figure 8—source data 1* and *2*).

## Blocking non-canonical NF-κB signalling and depletion of TNF together prevent inflammation caused by TRAF2 deficiency in keratinocytes

Deletion of one allele of NIK prevents lethality of *Traf2* knock-out (*Lin et al., 2011*; *Vallabhapurapu et al., 2008*). Together with the observation that TRAF2 knock-out keratinocytes also have high levels of NIK and the active form of NFκB2 (p52), we hypothesised that inhibiting non-canonical NF-κB signalling might prevent the inflammation in $Traf2^{EKO}$ mice. To test this, we crossed $Traf2^{EKO}$ with either *Map3k14 (Nik)$^{aly/aly}$* mice, that bear an inactive point mutant of NIK, or *Nfkb2* (p100)$^{-/-}$ mice. Contrary to our hypothesis, deficiency in *Map3k14* or *Nfkb2* did not prevent the psoriasis-like skin disease in $Traf2^{EKO}$ mice (*Figure 9* and *10*). Not only did $Traf2^{EKO}Map3k14^{aly/+}$ and $Traf2^{EKO}Map3k14^{aly/aly}$ mice develop skin inflammation, the onset of disease in

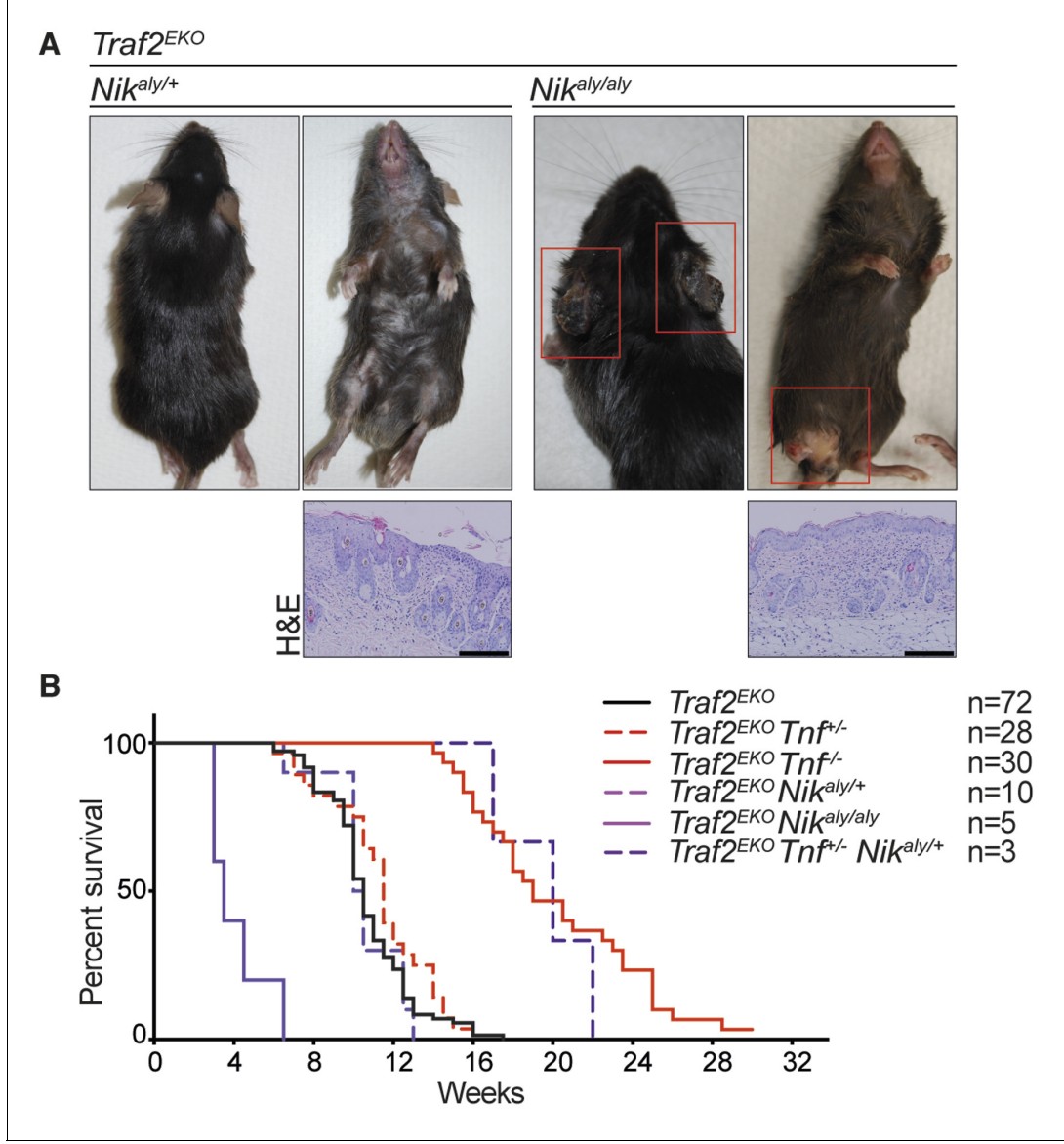

**Figure 9.** NIK mutation (*aly*) does not rescue the inflammation in $Traf2^{EKO}$ mice. (**A**) Representative images and skin sections of mice with indicated genotypes. Scale bars = 100 µm. (**B**) Kaplan-Meier graph indicating the time that the mice with the skin lesions, needed to be sacrificed according to animal ethics regulations.

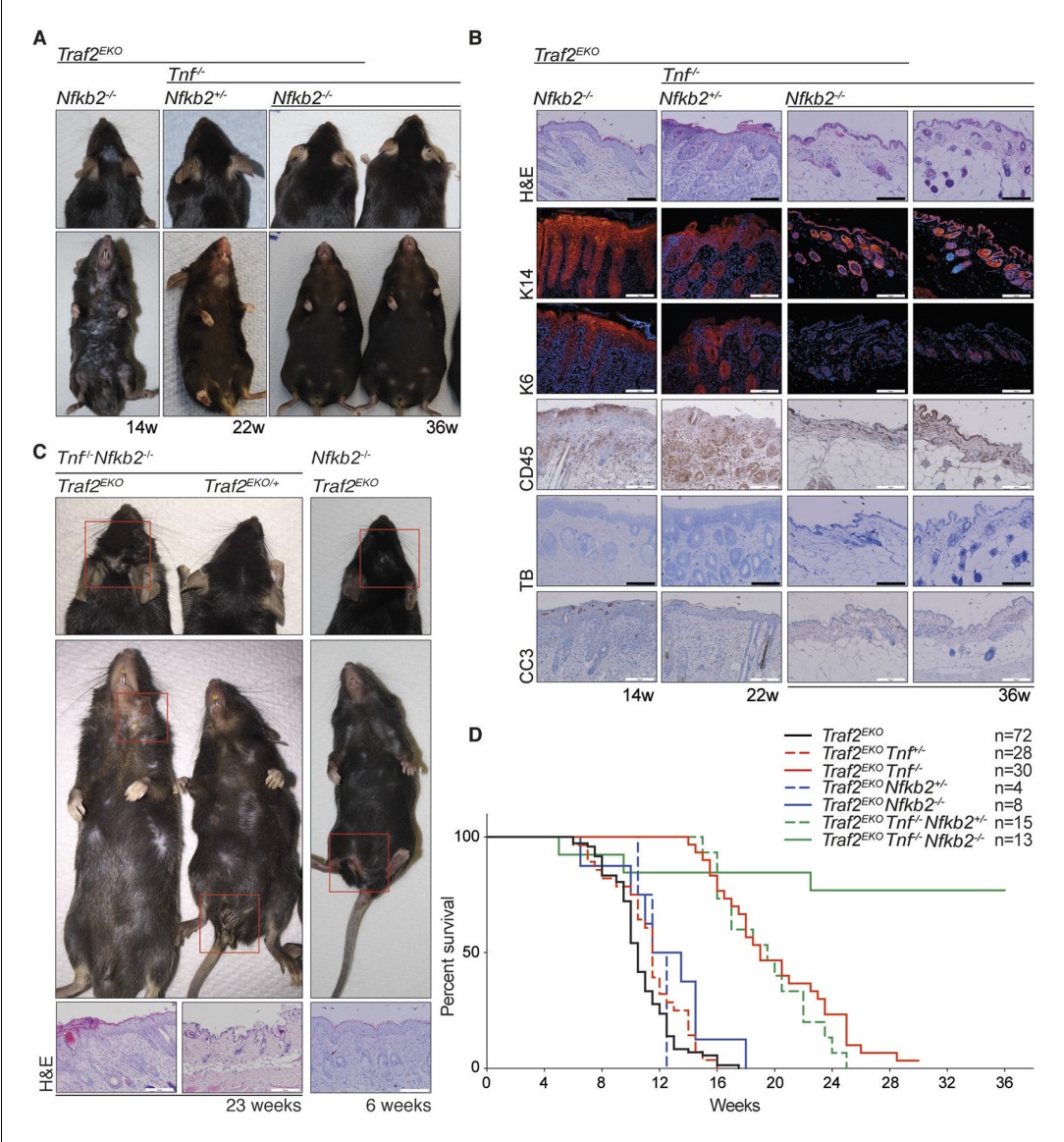

**Figure 10.** Depletion of both *Tnf* and *Nfkb2* rescues the skin inflammation caused by loss of TRAF2 in keratinocytes. (**A-C**) Representative images and skin sections of indicated mice stained with haematoxylin/eosin (H&E), immunostained for indicated epidermal differentiation markers (in red) and nuclei (Hoechst; in blue) and immunohistochemistry for pan leukocyte marker (CD45), cleaved caspase-3 (CC3) and toluidine blue (TB). Scale bars = 100 μm. (**D**) Kaplan-Meier graph depicting survival of indicated mouse strains.

*Traf2^{EKO}Map3k14^{aly/aly}* mice was earlier than in *Traf2^{EKO}* or *Traf2^{EKO}Map3k14^{aly/+}* mice (*Figure 9*). *Traf2^{EKO}Nfkb2^{-/-}* mice developed epidermal hyperplasia and skin inflammation in the same time frame as *Traf2^{EKO}* animals (*Figure 10*). Furthermore, the location of skin disease in both *Traf2^{EKO}-Map3k14^{aly/aly}* and *Traf2^{EKO}Nfkb2^{-/-}* was different to that in *Traf2^{EKO}* animals (*Figures 9A* and *10C*). Loss of *Traf2* usually caused lesions in the submental and forehead area, while lesions in *Traf2^{EKO}Map3k14^{aly/aly}* and *Traf2^{EKO}Nfkb2^{-/-}* mice appeared around the legs and sometimes spread over the whole ventral area (*Figures 9A*, *10A,C*). We were unable to examine *Traf2^{EKO}Tnf^{-/-}Map3k14^{aly/aly}* mice because *Tnf^{-/-}Map3k14^{aly/aly}* mice were highly susceptible to infections and difficult to breed (Data not shown). However, depletion of both TNF and NF-κB2 completely prevent the psoriasis-like inflammation in the *Traf2^{EKO}* mice (*Figure 10*). Three out of thirteen of the *Traf2^{EKO}Tnf^{-/-}Nfkb2^{-/-}* mice developed skin lesions at an early age around the mouth and anus (*Figure 10C,D*). This is likely to be due to the susceptibility of NFκB2 deficient mice to opportunistic infections because both

*Tnf*^-/-^*Nfkb2*^-/-^ and *Traf2*^EKO/EKO^*Nfkb2*^-/-^ mice also succumbed to such infections (*Figure 10C*) and (*Shinkura et al., 1996*; *Yin et al., 2001*). However, most of the *Traf2*^EKO^*Tnf*^-/-^*Nfkb2*^-/-^ mice were healthy up to 36 weeks of age without showing any sign of inflammation (*Figure 10*).

## Discussion

TRAF2 is believed to play a key role in TNF signalling, but this hypothesis has not been tested in all tissues, and there are conflicting data on its role. Some of the strongest evidence from knock-out and dominant negative TRAF2 transgenic mice suggests that loss of TRAF2 delays or reduces TNF-induced NF-κB and JNK and is required to protect cells from TNF-induced death (*Lee et al., 1997*; *Yeh et al., 1997*). How TRAF2 might perform these functions is not clear. On the one hand, a ΔRING TRAF2 is unable to protect *Traf2*^-/-^*Traf5*^-/-^ cells from TNF-induced death even though it is sufficient to reconstitute TNF-induced NF-κB (*Vince et al., 2009*; *Zhang et al., 2010*). This suggests that the E3 ligase activity of TRAF2 is not required for TNF-induced NF-κB but is required to protect cells from TNF-induced death. On the other hand, it has been demonstrated that the RING domain of TRAF2 is atypical and seemingly unable to recruit E2 ubiquitin-conjugating enzymes (*Yin et al., 2009*). A potential resolution for this conundrum was proposed by Alvarez et al. who suggested that the RING of TRAF2 required a lipid co-factor, S1P, to function as an E3 ligase (*Alvarez et al., 2010*). However, this solution is problematic because these authors showed that loss of SPHK1, and therefore S1P, prevented TNF-induced ubiquitylation of RIPK1 and TNF-induced activation of NF-κB; yet, neither of these two events depends upon the RING finger of TRAF2 (*Vince et al., 2009*; *Zhang et al., 2010*). Another difficulty with the Alvarez interpretation of their data is that *Sphk1*^-/-^ mice are viable and relatively normal (*Allende et al., 2004*), while *Traf2* knock-outs on a C57BL/6 background die before or shortly after birth (*Yeh et al., 1997*) and on a BALB/c background develop severe colitis and die within three weeks after birth (*Piao et al., 2011*). However, if their hypothesis is correct then the phenotype of the *Sphk1*-deficient mice should be equal to, if not more severe than, that of *Traf2*^-/-^ mice. Two groups have explored the implications of the Alvarez finding further: Adada et al. showed that SPHK1 is required for activation of MAPK p38 but not NF-κB after TNF stimulation (*Adada et al., 2013*), while Xiong et al. showed that neither SPHK1 nor SPHK2 are required for TNF-mediated activation of NF-κB and MAP kinases in macrophages (*Xiong et al., 2013*).

To explore this problem we performed a detailed analysis of TRAF2 function using cells from a number of different *Traf2*^-/-^ tissues and compared their responses to TNF with *Sphk1*^-/-^ cells. Surprisingly, *Traf2*- or *Sphk1*-deficient BMDMs activated NF-κB, JNK and ERK like wild-type cells in response to TNF, and were insensitive to TNF-induced cell death. However, *Traf2*^-/-^ cells were more sensitive than wild-type cells to Smac-mimetic-induced killing, showing that loss of TRAF2 increases the sensitivity of BMDMs to TNF if cIAPs are absent. There might be two reasons for this extra sensitivity: TRAF2 might recruit other E3 ligases that provide some protection in the absence of cIAPs from TNF-induced killing; alternatively, it has been reported that TRAF2 can K48-ubiquitylate caspase-8 and promote its proteosomal degradation, thereby inhibiting TRAIL-induced apoptosis (*Gonzalvez et al., 2012*). TRAF2 might therefore play a similar role downstream of TNFR1 in macrophages, leading to more TNF-induced death in the absence of cIAPs.

Because loss of TRAF2 had no noticeable effect on TNF signalling in macrophages, we were unable to compare the role of TRAF2 and SPHK1 in TNF signalling. Therefore, we extended our investigation into the role of TRAF2 and SPHK1 in MEFs, MDFs and keratinocytes. These cells required TRAF2 for full-strength TNF signalling and TRAF2 deficiency made them sensitive to TNF-induced death. However, SPHK1 deficiency did not affect either NF-κB or MAPK activation in these cells and they were resistant to TNF-induced death. Consistent with these data we found that ubiquitylation of RIPK1 upon TNF stimulation was normal in *Sphk1* knock-out MDFs but delayed and weakened in *Traf2*^-/-^ MDFs. TRAF2 has been shown to ubiquitylate the large subunit of caspase-8 and inhibit FasL or TRAIL-induced apoptotic cell death in human cell lines (*Gonzalvez et al., 2012*). However, we did not detect any caspase-8 ubiquitylation upon TNF stimulation in MDFs. These data indicate that role of TRAF2 in TNF signalling is cell type-specific. In cell types where TRAF2 is required for TNF signalling our data are in line with earlier studies on *Traf2*^-/-^ or dominant negative transgenic mice, because we see defects in JNK activation and delayed NF-κB activation (*Lee et al., 1997*; *Yeh et al., 1997*). However, in cell types where loss of TRAF2 results in defective TNF

signalling, we were unable to detect any equivalent defect in *Sphk1* deficient cells, arguing against the idea that S1P is an essential cofactor required for TNF-induced, TRAF2-dependent, NF-κB.

Studies on TRAF2 deficient B cells revealed a crucial role for TRAF2 in B cell development and examined the consequences of *Traf2* deficiency on CD40 and BAFF signalling (*Gardam et al., 2008*; *Hostager et al., 2003*). Aside from lymphocytes, however, a physiological role for TRAF2 in cells or tissues has not been well defined. Recent studies suggest that TRAF2 could have a crucial role in adult tissue homeostasis; three-week-old *Traf2*$^{-/-}$ mice on the BALB/c background developed spontaneous colitis, and this phenotype was due to the death of the colonic epithelium (*Piao et al., 2011*). In another study, intraperitoneal injection of Tamoxifen to *Traf2*$^{loxp/loxp}$*Rosa-Cre-ER*$^{T2}$ mice induced inflammation in the intestine and mortality after one week (*Petersen et al., 2015*). However, it is not entirely clear whether these defects are due to the loss of TRAF2 in epithelial cells or hematopoietic cells. Therefore, we specifically targeted *Traf2* in keratinocytes and investigated the role of TRAF2 in the epidermis. Keratinocyte-specific *Traf2* knock-out mice developed epidermal hyperplasia and severe skin inflammation, leading to noticeable lesions around head and submental area by 6 weeks of age. However, consistent with previous reports (*Allende et al., 2004*; *Xiong et al., 2013*), no sign of inflammation was detected in the skin of *Sphk1*$^{f/f}$*Sphk2*$^{-/-}$*Rosa-Cre-ER*$^{T2}$ mice following activation of Cre-mediated gene deletion with Tamoxifen (Data not shown). Thus, total lack of SPHK isoenzymes did not phenocopy the skin defects seen in the absence of TRAF2, again arguing against an essential role for S1P in TRAF2's function.

Several mouse models of skin inflammation have been rescued by depletion of TNF (*Gerlach et al., 2011*; *Nenci et al., 2006*; *Pasparakis et al., 2002*). While TNF is a major inflammatory mediator and directly drives production of a host of inflammatory cytokines through TNFR1 signalling, it has been proposed that TNF-induced cell death, specifically necroptosis, might also generate Damage Associated Molecular Patterns (DAMPs) that could activate Toll Like Receptors, or other analogous receptors to indirectly promote inflammatory cytokine production (*Gerlach et al., 2011*; *Pasparakis and Vandenabeele, 2015*). The relative contribution of these two pathways to inflammation in vivo, and particularly in human pathology, is still unclear. Keratinocyte hyperplasia in *Sharpin*$^{cpdm/cpdm}$ mice can be driven by TNF-induced apoptotic death, because deletion of just one allele of *Casp8* or epidermal *Fadd* deletion markedly ameliorated or prevented the TNFR1-dependent inflammatory phenotype (*Kumari et al., 2014*; *Rickard et al., 2014a*). Furthermore, epidermal *cFlip (cflar)* knock-out causes excessive apoptotic death and an initial wave of very rapid hyperplasia in the epidermis that is rescued by TNF inhibition (*Panayotova-Dimitrova et al., 2013*). Complicating this picture, caspase-8 deficiency in keratinocytes also causes inflammation in the skin and this is rescued partially by *Tnf* or *Tnfr1* deletion (*Kovalenko et al., 2009*). However, in this case the inflammation was attributed to constitutive activation of Interferon Regulatory Factor-3 (IRF3), which induces the expression of cytokines and chemokines in the epidermis of *casp8* epidermal knock-out independently of TNF (*Kovalenko et al., 2009*). The skin inflammation of epidermal caspase-8-deficient mice (*Weinlich et al., 2013*), but not of epidermal *cFlip*-deficient mice (*Panayotova-Dimitrova et al., 2013*), is completely rescued by loss of RIPK3. Therefore, Weinlich et al. suggested that necroptosis is the cause of inflammation in the epidermal *casp8* knock-out. Similarly, *Fadd*$^{EKO}$, *Ripk1*$^{EKO}$ and *Ripk1*$^{-/-}$ mice develop skin inflammation that has been attributed to RIPK3-dependent necroptosis, highlighting the importance of this pathway in the skin (*Bonnet et al., 2011*; *Dannappel et al., 2014*; *Dillon et al., 2014*; *Rickard et al., 2014b*). In our model, TNF deficiency only delayed the *Traf2*$^{EKO}$ skin phenotype and did not rescue it completely, and inflammation induced by loss of TRAF2 was not prevented by loss of the necroptotic effector MLKL. This was consistent with our analyses on the type of TNF-induced cell death in *Traf2*$^{-/-}$ keratinocytes, which was predominantly apoptotic. Therefore, we have crossed *Traf2*$^{EKO}$ to *Mlkl*$^{-/-}$*Casp8*$^{-/-}$ mice and rescued the early TNF-dependent skin inflammation. Unfortunately, we were unable to age these mice more than 15 weeks, as they developed lymphadenopathy, and we could therefore not examine the effect of *Mlkl*$^{-/-}$*Casp8*$^{-/-}$ on the late onset inflammation in the skin. Recently, it has been shown that apoptotic cells could also release DAMPs such as ATP (*Poon et al., 2014*), and our results support the idea that excessive apoptosis can be inflammatory.

Although the microscopic phenotypes of *Traf2*$^{EKO}$ or *Traf2*$^{EKO}$*Tnf*$^{-/-}$ were similar to human psoriatic lesions, we are not sure whether the infiltration of the immune cells entirely resembles human psoriasis. In our models we observed infiltration of neutrophils and IFNγ-producing CD4$^+$ T cells to the skin, which is similar to human psoriasis. However, dermal dendritic cells, macrophages, other

subtypes of T helper cells (Th2 and Th17) and cytotoxic T cells have also been reported to have essential roles in initiation and development of psoriasis (*Nestle et al., 2009*). More investigation about the involvement of different immune cells in different stages of the disease is clearly warranted.

While *Tnf* deficiency delayed the appearance of the inflammatory skin phenotype in *Traf2*$^{EKO}$ mice, at later time points the epidermis thickened and there was a leukocyte infiltration. Because *Traf2* deficiency is specific to keratinocytes this implies that there are one or more keratinocyte-intrinsic factors that either promote proliferation and attract inflammatory cells to the skin or attract inflammatory cells which promote keratinocyte proliferation. One possibility is that the constitutive non-canonical NF-κB signalling observed in *Traf2*$^{EKO}$*Tnf*$^{-/-}$ keratinocytes promotes their proliferation. However, in contrast to *Traf2*$^{-/-}$ lymphocytes (*Gardam et al., 2008*), proliferation of *Traf2*$^{EKO}$*Tnf*$^{-/-}$ keratinocytes in culture was not altered compared to controls. This suggests that epidermal hyperplasia in the *Traf2*$^{EKO}$*Tnf*$^{-/-}$ mice is not the result of an intrinsic growth advantage of *Traf2*$^{-/-}$ keratinocytes. IL-17 plays an important role in inflammatory diseases and monoclonal antibodies targeting IL-17 such as secukinumab and ixekizumab are performing well in psoriasis patients (*Tse, 2013*). These successes indicate that IL-17 plays an important role in human psoriasis and it has been shown that IL-17 can induce proliferation of keratinocytes in vivo (*Ha et al., 2014*). TRAF3 interacting protein 2 and TRAF3 have been linked to IL-17 signalling (*Qian et al., 2007*; *Zhu et al., 2010*) where TRAF3 negatively regulates IL-17 receptor signalling. We therefore considered that loss of TRAF2 might have a similar effect to loss of TRAF3 and that IL-17 might promote keratinocyte proliferation in TRAF2-deficient keratinocytes. However, in our experiments, IL-17 did not increase *Traf2*$^{EKO/EKO}$*Tnf*$^{-/-}$ keratinocyte proliferation compared to *Traf2*$^{EKO/+}$*Tnf*$^{-/-}$ cells in vitro and loss of TRAF2 had no impact on IL-17-mediated activation of NF-κB and p38 in keratinocytes. While this does not exclude a role for IL-17 in the psoriasis-like skin phenotype observed in *Traf2*$^{EKO}$ or *Traf2*$^{EKO}$*Tnf*$^{-/-}$ mice, we hypothesized that *Traf2*$^{EKO}$*Tnf*$^{-/-}$ keratinocytes produced other factors to recruit inflammatory cells to the skin.

We therefore analysed and compared supernatants from *Traf2*$^{EKO/EKO}$*Tnf*$^{-/-}$ and *Traf2*$^{EKO/+}$*Tnf*$^{-/-}$ keratinocytes using Mass Spectrometry, Western blot and ELISA. However, we were unable to detect cytokines such as IL-6, IL-23, IL-24, IL-33, TSLP, IL-18, MCP-1 and GM-CSF using these methods (Data not shown). This failure might be due to low levels of secretion of these factors by keratinocytes maintained in serum-free media, so we investigated the transcription of inflammatory genes. At the mRNA level, several inflammatory genes were highly elevated in *Traf2*$^{EKO/EKO}$*Tnf*$^{-/-}$ compared to *Traf2*$^{EKO/+}$*Tnf*$^{-/-}$ keratinocytes, and many of these, such as M-CSF and IL-23, have been implicated in the development of inflammatory diseases (*Nair et al., 2009*; *van Nieuwenhuijze et al., 2013*; *Yoshiki et al., 2014*).

Depletion of non-canonical NF-κB signalling prevents early lethality in *Traf2*$^{-/-}$ mice (*Lin et al., 2011*; *Vallabhapurapu et al., 2008*) but did not rescue the *Traf2*$^{EKO}$ skin phenotype in our model. Surprisingly, the inflammatory lesions appeared with a different pattern in *Traf2*$^{EKO}$*Nik*$^{aly/aly}$ and the onset of inflammation was even earlier than in *Traf2*$^{EKO}$ mice. This could be either due to the susceptibility of *Nik*$^{aly/aly}$ to autoimmunity that is exacerbated by the loss of TRAF2 or there could be another unknown pathway inhibited by TRAF2 that is worsened by NIK mutation. In *Traf2*$^{EKO}$*Nfkb2*$^{-/-}$ animals, the onset of inflammation was similar to *Traf2*$^{EKO}$ mice, but the development of disease was slightly slower and mice survived marginally longer.

These data show that TNF-dependent cell death, but not non-canonical NF-κB, is a driver of the early inflammation in *Traf2*$^{EKO}$ mice. Constitutive activation of non-canonical NF-κB in *Traf2*$^{EKO}$*Tnf*$^{-/-}$ mice might, however, be responsible for the later onset skin disease that developed in these *Tnf*-deficient mice (*Figure 11*). To test this, we deleted both *Tnf* and *Nfkb2*, which completely rescued the skin inflammation in *Traf2*$^{EKO}$ mice. These data provide insights into the mechanism of initiation and development of inflammatory skin disease that may be important to further understanding of diseases such as psoriasis. Moreover, our findings provide further evidence that TNF-induced apoptosis can play a role in inflammatory skin disease and indicate that defects in keratinocytes themselves can initiate TNF-dependent and -independent inflammation. Importantly, we have shown that non-canonical NF-κB signalling can be the driver of inflammation in epithelial cells. This observation might be relevant to human psoriasis because it has been reported that the levels of TWEAK and its receptor Fn-14 are elevated in human psoriatic lesions (*Cheng et al., 2015*), and it is well established that TWEAK/Fn14 signalling induces degradation of TRAF2 and activation of non-canonical NF-κB signalling (*Saitoh et al., 2003*; *Vince et al., 2008*). Thus, our results suggest that inhibition of non-

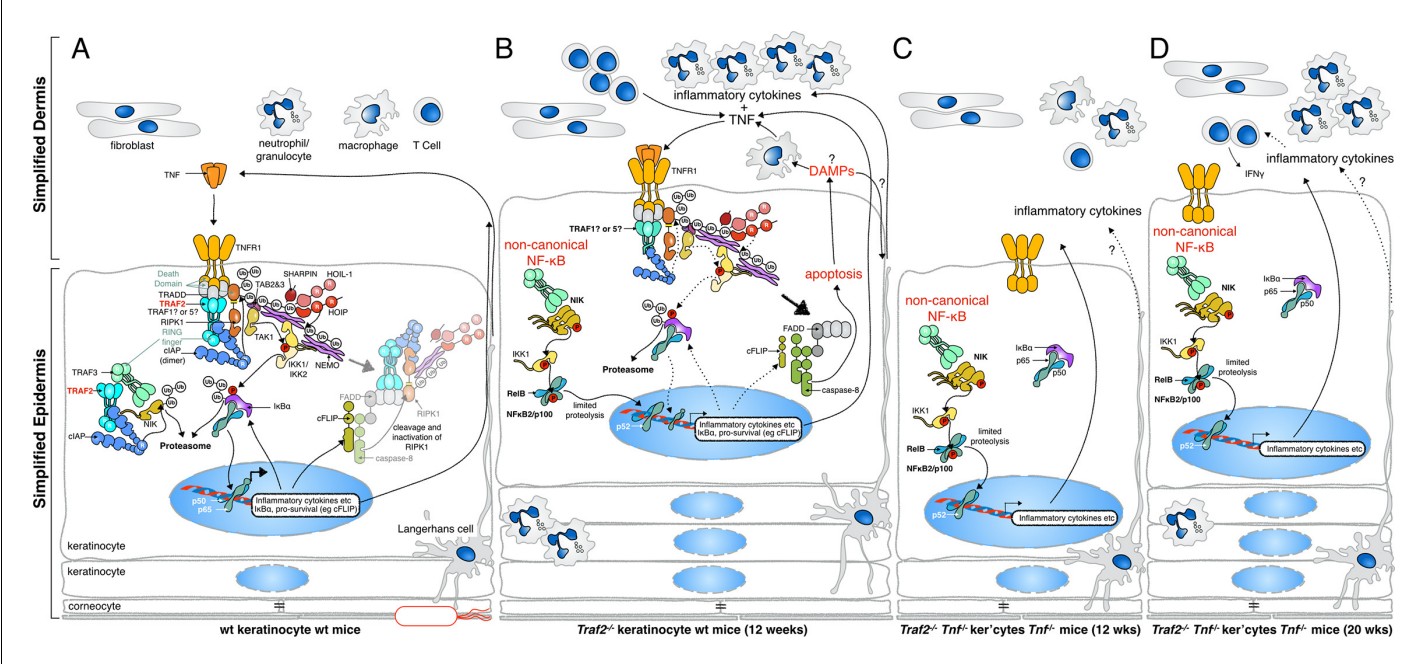

**Figure 11.** Proposed mechanisms by which loss of TRAF2 in keratinocytes causes a psoriasis-like skin inflammation. (**A**) In normal wild-type skin TNF signalling might be activated by local penetration of bacteria into the epidermis. This will induce TNF, presumably by keratinocytes themselves or other resident cells such as Langerhans cells, and promotes canonical NF-κB signalling, inflammatory cytokine production and expression of prosurvival genes. If this response nullifies the threat, TNF signalling is turned off and homeostasis of the skin is maintained. (**B**) In the absence of TRAF2 in keratinocytes, TNF production can induce apoptotic cell death. This cell death, possibly because of the release of DAMPs, recruits neutrophils and other inflammatory cells to the skin and causes epidermal hyperplasia. Loss of TRAF2 also causes constitutive non-canonical NF-κB activation in viable keratinocytes, which increases the production and release of inflammatory cytokines, including TNF. Thus, TRAF2-deficient keratinocytes do not need to be stimulated by bacteria to produce TNF, and TNF-induced death sets up a potentially vicious cycle of inflammation. (**C**) Concomitant deletion of *Tnf* in mice with *Traf2*-deficient keratinocytes breaks this vicious cycle and prevents apoptosis of keratinocytes and early onset of the psoriasis-like phenotype. (**D**) In the absence of TNF, *Traf2*-deficient keratinocytes still produce inflammatory cytokines via the non-canonical NF-κB pathway that ultimately generate the same psoriasis-like phenotype in mice but with slower onset. These cytokines recruit inflammatory immune cells to the skin, including neutrophils and IFNγ+ T cells.

canonical NF-κB signalling might be beneficial in chronic inflammatory disease together with TNF inhibitors.

## Materials and methods

### Mice

Mice were maintained at the Walter and Eliza Hall Institute (WEHI) under the approval of WEHI ethics committee. Animal technicians performed unbiased mouse monitoring and informed us if mice developed skin inflammation and needed to be sacrificed according to the WEHI animal ethic guidelines.

### Immunobloting

Cell lysates were prepared using DISC buffer (1% NP-40, 10% glycerol, 150 mM NaCl, 20 mM Tris pH 7.5, 2 mM EDTA, Roche [Basel, Switzerland] cOmplete protease inhibitor cocktail, 2 mM sodium orthovanadate, 10 mM sodium fluoride, β-glycerophosphate, $N_2O_2PO_7$). Cell lysates were loaded on NuPAGE Bis-Tris gels (Life Technologies, Carlsbad, CA) and transferred on to Immobilon-P PVDF membranes (Millipore, Billerica, MA) or Hybond-C Extra (GE Healthcare, Little Chalfont, UKGE Healthcare). Membranes were blocked and antibodies diluted in 5% skim milk powder or Bovine Serum Albumin (BSA) in 0.1% Tween20 in PBS or TBS. Antibodies used for Western blot were: Cleaved caspase-3 (9661) and -8 (8592), phospho-JNK1/2 (4668P), phospho-p38 (4511), p38 (9212),

caspase-8 (4927), JNK1/2 (9252), IκBα (CN: 9242) and phospho-p65 (3033) from Cell Signaling Technology (Danvers, MA), $\beta$-actin (A-1978; Sigma-Aldrich, St. Louis, MO), cFLIP (AG-20B-0005; Adipogen, Epalinges, Switzerland), RIPK1 (610458; BD Transduction Laboratories, San Jose, CA) and VDAC1 (AB10547; Millipore). Monoclonal antibody for TRAF2 was made in-house and raised against the RING domain of TRAF2 (Clone: 6/12-2D1-22-1). cIAP1 antibody was made in house and is distributed by ENZO Life Sciences (1E1-1-10; Farmingdale, NY). MLKL antibody was made in-house as described (*Murphy et al., 2013*). Signals were detected by chemiluminescence (Millipore) after incubation with secondary antibodies conjugated to HRP.

## Immuno precipitation of ubiquitin-conjugates (TUBE IP)

Tandem Ubiquitin Binding Entities (TUBE1: binds to K6-, K11-, K48- and K63-linked polyubiquitin; Lifesensors, Malvern, PA) beads were used to purify ubiquitin conjugates from MDFs according to the manufacturer's recommendations (*Hjerpe et al., 2009*). Wild–type, *Traf2*[-/-] and *Sphk1*[-/-] immortalised MDFs were treated with TNF (100 ng/ml) in the absence or presence of Q-VD-OPh (QVD; 10μM; SM Biochemicals, Anaheim, CA) and Necrostatin (Nec; 50μM; Biomol, Farmingdale, NY) for the times indicated and lysed in ice-cold DISC buffer. The lysates were incubated with TUBE beads for 2 hr at 4°C before washing 3x in ice-cold PBS-Tween. Washed beads were resuspended in 1x SDS sample buffer, separated on SDS-PAGE gels and subjected to Western blotting and probed with the indicated antibodies.

## Histology and immunofluorescence

Skin samples were fixed in 10% neutral buffered formalin, paraffin-embedded, and sectioned for routine histology staining (H&E). For skin immunofluorescence, parafin sections were de-waxed, subjected to heat-induced epitope retrieval (HIER) with citrate buffer, blocked in 3% goat serum, then permeabilised with 0.3% Triton X-100, and stained with keratin 6 (Covance) or keratin 14, and goat anti-rabbit Alexa 594 (Invitrogen, Carlsbad, CA) antibodies. Nuclei were visualized using Hoechst (Invitrogen). For tissue immunohistochemistry, sections were dewaxed and subjected to antigen retrieval in trypsin buffer or HIER with citrate buffer and stained with anti-CC3 (9691; Cell Signaling Technology, Danvers, MA), anti-CD45 (BD, San Jose, CA) and goat anti-rabbit biotinylated antibodies. Images were taken using a DP72 microscope and cellSens Standard software (Olympus, Tokyo, Japan).

## Isolation of cells from skin and cytokine measurement

Skin was separated from the tail of an adult mouse and incubated in dispase II (20 mg/ml) at 4°C for 24 hr and keratinocytes and MDFs were isolated from the epidermal and dermal layers respectively as described before (*Etemadi et al., 2013*; *Rickard et al., 2014a*). TNF ELISA was performed using (88–7324; eBioscience, San Diego, CA) kit following the manufacturer's protocol.

## Generation of immortalised MEFs and MDFs

MEFs were generated from E15 embryos by removing their head and liver and passing through a cell strainer. MEFs and MDFs were immortalised by infecting these cells with a lentivirus expressing SV40 Large T antigen.

## qPCR inflammation array

Cultured keratinocytes in one well of a 6-well plate were lysed in 500 μl TRIzol (Invitrogen) and the RNA was purified using PureLink RNA purification kit (Ambion, Carlsbad, CA). cDNA was made with SuperScript VILO cDNA Synthesis kit (Invitrogen) and qPCR was performed using TaqMan Open Array mouse inflammation panel (Applied Biosystems, Carlsbad, CA) on a QuantStudio 12K Flex machine (Life Technologies) following the manufacturer protocols.

## Statistical analyses of qPCR inflammation array data

Ct values were exported from Life Science's Expression Suite software and statistical analysis was undertaken using the limma software package (*Ritchie et al., 2015*). The maximum measurable Ct value was 40, so Ct values were transformed to a log2 expression scale by subtracting the Ct values from 41. The expression values were normalized using cyclic loess normalization (*Bolstad et al., 2003*)

with house-keeping probes up-weighted 10-fold. The cyclic method was set to 'affy', the loess span was 0.7 and 3 cyclic iterations were used. The probes Gapdh, Pgk, Hmbs, Hprt and Ppia are treated as house-keeping. Probes were filtered out as unexpressed if they failed to achieve a normalized value of 4 in at least 3 samples. Comparisons were made between $Traf2^{-/-}Tnf^{-/-}$ and $Traf2^{+/-}Tnf^{-/-}$ keratinocytes using empirical Bayes t-statistics (*Smyth, 2004*). The empirical Bayes hyper-parameters were estimated robustly, including an abundance-dependent trend. The false discovery rate (FDR) was controlled below 0.05 using the method of Benjamini and Hochberg (*Benjamini and Hochberg, 1995*).

### Cell death and viability assay

Cells were harvested after 24-hr treatment with human Fc-TNF (made in-house, referred as 'TNF' in the manuscript) and Smac-mimetic (TetraLogic), and cell death was measured by Propidium Iodide (PI) staining and flow cytometry performed on a FACSCalibur flow cytometer (BD Biosciences, San Jose, CA) and analysed using Weasel (WEHI). Keratinocyte viability was assayed using CellTiter 96 (Promega, Madison, WI) according to the manufacturer's instructions.

### Skin digestion and flow cytometry

Epidermal and dermal layers of ears were separated after 30 min incubation with 2.4 U/ml Dispase (GIBCO, Carlsbad, CA) at 37°C, followed by cutting into fine pieces and digested with 50 U/ml collagenase for 45 min at 37°C. Single-cell suspension was made in FACS buffer (PBS, 0.5% BSA; Sigma-Aldrich) and blocked with Fc block (eBioscience) and stained with the following antibodies in FACS buffer at 4°C as described before (*Chopin et al., 2013*). Antibodies raised against CD11c (N418), CD45.2 (A20), MHCII (M514.15.2), CD8α (53–6.7), CD4 (GK1.4), IFNγ, Ly6G (1A8), were purchased from BD biosciences. TCRβ (H57-597) and cd11b (M1/70) were purchased from eBioscience.

### T cell stimulation

Dermal T cells were isolated from the skin and resuspended in RPMI-1640 supplemented with 10% heat-inactivated fetal calf serum, 2 mM L-Glutamine (GIBCO), 50 µM 2-mercaptoethanol (Sigma, St. Louis, MO) 100 U/ml penicillin/streptomycin (GIBCO). Cells were stimulated for 4 hr in presence of PMA (20 ng/ml) and Ionomycin (1 µg/ml) in presence of the GolgiPlug (BD Biosciences).

### Reconstitution experiments

Mice were irradiated with 550 rads twice with 3 hr minimum between irradiation. Approximately 5 million cells from the bone marrow of C57/BL6 or IFNγ mice were injected by tail i.v. to $Traf2^{EKO}$. Mice were maintained on 2 mg/ml neomycin in drinking water for three weeks post irradiation.

### Fractionation and Blue-Native PAGE

Stimulated keratinocytes were permeabilised in buffer containing 0.025% digitonin. Cytosolic and crude membrane fractions were further solubilized in 1% digitonin, resolved by Bis·Tris Native PAGE as described before (*Hildebrand et al., 2014*), and immuno-probed for MLKL.

## Acknowledgements

We thank staff in the WEHI Bioservices facility, Heinrich Korner for $Tnf^{-/-}$ mice, James Murphy and Warren Alexander for $Ifng^{-/-}$ and $Mlkl^{-/-}$, Robert Brink for $Traf2^{lox/lox}$ and Roland Schmid for $Nfkb2^{-/-}$ mice. This work was supported by NHMRC grants (1025594, 1046984, 1048278, 546272), NHMRC fellowships to JS (541901, 1058190), SLN (1058238), NIH grants (HL67330 and CA77839 TH), SNSF project grant (310030-138085) and the Thomas William Francis and Violet Coles Trust Fund with additional Victorian State Government Operational Infrastructure Support, NHMRC IRIISS grant (361646). We thank George Varigos for discussions and support.

# Additional information

## Funding

| Funder | Grant reference number | Author |
|---|---|---|
| Victorian State Government Operational Infrastructure Support | | Nima Etemadi<br>Michael Chopin<br>Holly Anderton<br>Maria C Tanzer<br>James A Rickard<br>Waruni Abeysekera<br>Cathrine Hall<br>Sukhdeep K Spall<br>Gordon K Smyth<br>David L Vaux<br>Stephen L Nutt<br>Ueli Nachbur<br>John Silke |
| Independent Research Institutes Infrastructure Support | 361646 | Nima Etemadi<br>Michael Chopin<br>Holly Anderton<br>Maria C Tanzer<br>James A Rickard<br>Waruni Abeysekera<br>Cathrine Hall<br>Sukhdeep K Spall<br>Gordon K Smyth<br>David L Vaux<br>Stephen L Nutt<br>Ueli Nachbur<br>John Silke |
| Thomas William Francis and Violet Coles Trust Fund | | James A Rickard<br>John Silke |
| National Health and Medical Research Council | 546272 | Bing Wang |
| National Institutes of Health | HL67330 | Yuquan Xiong<br>Timothy Hla |
| National Institutes of Health | CA77839 TH | Yuquan Xiong |
| Schweizerischer Nationalfonds zur Förderung der Wissenschaftlichen Forschung | 310030-138085 | Wendy Wei-Lynn Wong |
| National Health and Medical Research Council | 1048278 | Stephen L Nutt |
| National Health and Medical Research Council | 1058238 | Stephen L Nutt |
| Schweizerischer Nationalfonds zur Förderung der Wissenschaftlichen Forschung | PA00P3_126249 | Ueli Nachbur |
| National Health and Medical Research Council | 1058190 | John Silke |
| National Health and Medical Research Council | 541901 | John Silke |
| National Health and Medical Research Council | 1025594 | John Silke |
| National Health and Medical Research Council | 1046984 | John Silke |

The funders had no role in study design, data collection and interpretation, or the decision to submit the work for publication.

## Author contributions

NE, MC, UN, Conception and design, Acquisition of data, Analysis and interpretation of data, Drafting or revising the article; HA, CH, SKS, Designed and performed experiments, Acquisition of data, Analysis and interpretation of data; MCT, JAR, Designed and performed experiments, Acquisition of data, Analysis and interpretation of data, Drafting or revising the article; WA, Analyzed data, provided helpful suggestions and discussions, Analysis and interpretation of data, Drafting or revising the article; BW, Acquisition of data, Analysis and interpretation of data; YX, TH, Analyzed data, Provided helpful suggestions and discussions, Acquisition of data, Drafting or revising the article; SMP, CSB, Analyzed data, Acquisition of data, Drafting or revising the article; WW-LW, Analyzed data, Drafting or revising the article, Contributed unpublished essential data or reagents; ME, Had significant input on conception and design of the revised manuscript after the first submission, Conception and design, Drafting or revising the article; GKS, Analysis and interpretation of data, Drafting or revising the article; DLV, SLN, Conception and design, Analysis and interpretation of data, Drafting or revising the article; JS, Analyzed data, Provided helpful suggestions and discussions, Conception and design, Analysis and interpretation of data, Drafting or revising the article

## Author ORCIDs

Nima Etemadi, http://orcid.org/0000-0002-7137-1373
Gordon K Smyth, http://orcid.org/0000-0001-9221-2892
Stephen L Nutt, http://orcid.org/0000-0002-0020-6637

## Ethics

Animal experimentation: Mice were maintained at the Walter and Eliza Hall Institute (WEHI) under the approval of WEHI ethics committee and Institute guidelines. All procedures were specifcally approved under WEHI Ethics Project Number 2011.013 and 2014.015.

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
