## [Decision Letter]

Thank you for submitting your work entitled "TRAF2 regulates TNF and NF-κB signalling to suppress apoptosis and skin inflammation independently of SPHK1" for peer review at *eLife*. Your submission has been favorably evaluated by Tadatsugu Taniguchi (Senior Editor) and three reviewers, one of whom is a member of our Board of Reviewing Editors.

The following individuals responsible for the peer review of your submission have agreed to reveal their identity: Manolis Pasparakis and Hao Wu (peer reviewers).

The reviewers have discussed the reviews with one another and the Reviewing editor has drafted this decision to help you prepare a revised submission.

The authors used genetic, cellular, immunological, and histological tools to investigate the roles of TRAF2 in inflammation and cell death. TRAF2 knockout does not affect TNF-induced canonical NF-κB or MAPK signaling, or cell death in macrophages. The authors show that TRAF2 is required for TNF signaling in fibroblasts and keratinocytes, but SPHK1 (sphingosine kinase 1) knockout cells did not show altered canonical NF-κB signaling, MAPK, or apoptotic cell death when compared to wild-type cells. Loss of TRAF2 leads to epidermal hyperplasia and psoriasis-like inflammation in mice, which can be delayed but not fully reversed by deletion of TNF. Deletion of TRAF2 activates the non-canonical NF-κB signaling pathway through NIK, and double deletion of NF-κB2 and TNF fully rescued inflammation induced by TRAF2 deletion. In addition, they conclude that in contrast to its previously suggested essential in regulating TRAF2 E3 ligase activity dependent TNF signalling, SPHK1 is not required for TNF-induced activation of NF-κB or MAPK and for resistance to TNF-induced cell death.

This is a very nice manuscript reporting clear and well-controlled experiments that reveal an important function of TRAF2 in the maintenance of skin homeostasis. Also, the data on the (lack of) function of SPHK1 in TNF signalling is very convincing. The authors should discuss more carefully the skin pathology developing in their mice comparing with human psoriasis in terms of histological findings, the immune infiltrates, gene expression as well as the genetic studies.

Reviewer #1 (Minor Comments):

1) The Materials and methods is missing information about how TUBE experiment was conducted, and source of RIPK1 and cIAP antibodies.

2) Figure 2 is missing blot for levels of anti ERK, JNK and p38 (A) and anti JNK (E).

3) The authors should be more consistent in all blots regarding time points when they compare effects of WT, *Traf2^-/-^* and *Sphk1^-/-^* cells.

4) In Figure 3, why is there no observed caspase-8 ubiquitination in WT and *Sphk1^-/-^* cells upon TNF stimulation, since cIAPs and TRAF2 (Gonzalvez et al., 2012) should be active? The authors may consider discussing this issue.

5) Figure 3: the authors should include also TUBE IP experiment with longer time points for *Traf2^-/-^* which would confirm their hypothesis about delayed NF-kB activation.

6) Figure 6: the authors should add FACS contour plot and corresponding bar diagram for CD11b+Ly6C+ infiltrating neutrophils in *Traf2^EKO^Tnf^-/-^* as they did for other cell suspensions, and compare the data with results for CD4^+^ T cells.

7) The authors should focus in Discussion on experiments with IL-17 and discuss potential other signals for activation of non-canonical NF-kB independent of TNF.

8) Figure 10: Kaplan-Meier graph does not depict final result with a survival of *Traf2^EKO^Tnf^-/-^Nfkb2^-/-^* mice for up to 36 weeks. Instead, it is showing only survival percentage for other 6 genotypes. Since this survival curve is one of the most crucial results, the authors should add it.

9) The authors describe that loss of *Traf2* mostly induced lesions in anterior parts of the body, while animals lacking active TRAF2 and functional non-canonical NF-kB signaling were more susceptible to develop lesions in ventral and posterior parts. They should discuss possible reasons for these differences.

10) In the last Results paragraph, the authors twice talk about the phenotype of *Traf2^EKO^*/+*Tnf^-/-^Nfkb2^-/-^* which isn't presented nowhere on figures. Please, provide figures showing these results or correct that paragraph.

Reviewer #2 (Minor Comments):

In Figure 1 and Figure 2, the β actin blots are not equalized among the lanes. It probably will not change the conclusions, but loading the same amount of samples to each lane will make the figures look more convincing.

Reviewer #3 (Minor Comments):

The authors state that the triple mutant mice (TRAF2EKO, *Tnf^-/-^, Nfkb2^-/-^*) 'occasionally' developed skin lesions but do not indicate how many mice have been analysed and how many developed lesions. Please clearly state the number of mice analysed for each genotype. It would also be helpful to clearly state the age of mice shown in all figures.

There is a wrong citation in the Discussion "Several mouse models of skin inflammation have been rescued by depletion of TNF (Gerlach et al., 2011; Nenci et al., 2006; Pasparakis, Schmidt-Supprian, and Rajewsky, 2002)." The citation Pasparakis, Schmidt-Supprian, and Rajewsky, 2002 is not the right one, the correct one is Pasparakis, Courtois et al., 2002.

Figure 6 in the subsection “Psoriasis-like inflammation in *Traf2^EKO^* mice is characterised by skin infiltration of neutrophils and IFNγ producing CD4^+^ T cells in” referring to the data on IFNγ+ T cells should be Figure 6.

[Editors' note: further revisions were requested prior to acceptance, as described below.]

Thank you for resubmitting your work entitled "TRAF2 regulates TNF and NF-κB signalling to suppress apoptosis and skin inflammation independently of Sphingosine Kinase 1" for further consideration at *eLife*. Your revised article has been favorably evaluated by Tadatsugu Taniguchi (Senior Editor), a Reviewing Editor, and one reviewer. The manuscript has been improved but there are some remaining issues that need to be addressed before acceptance, as outlined below.

The inflammatory skin lesions developing in mice with keratinocyte-specific TRAF2 knockout can be described as psoriasis-like, but it is not correct to write that these mice develop psoriasis, as in the following instances in the manuscript:

In the subsection “Psoriasis-like inflammation in *Traf2^EKO^* mice is characterised by skin infiltration of neutrophils and IFNγ producing CD4^+^ T cells in”: "However, both wild type and *Ifng^-/-^*chimeras developed skin inflammation to the same extent indicating that IFNγ is not a major contributor to the psoriasis (Figure 7)."

In the subsection “Blocking non-canonical NF-κB signalling and depletion of TNF together prevent inflammation caused by TRAF2 deficiency in keratinocytes”: "Contrary to our hypothesis, deficiency in *Map3k14* or *Nfkb2* did not prevent the psoriasis in *Traf2^EKO^* mice (Figure 9 and Figure 10)."

Discussion: "Unfortunately, we were unable to age these mice more than 15 weeks as they developed lymphadenopathy and could therefore not examine the effect of *Mlkl^-/-^Casp8^-/-^* on the late onset psoriasis."

Discussion: "While this does not exclude a role for IL-17 in the psoriasis observed in *Traf2^EKO^* or *Traf2^EKO^Tnf^-/-^* mice we hypothesized that *Traf2^EKO^Tnf^-/-^* keratinocytes produced other factors to recruit inflammatory cells to the skin."

In Figure 11, the schematic model gives the impression that in the absence of TNF the inflammatory skin lesions are induced by M-CSF and IL-23 released by keratinocytes and IFNγ produced by T cells. As there is no data in the paper supporting causal function of these cytokines, it would be more appropriate not to refer to specific cytokines or to at least clearly describe in the scheme and the legend that these cytokines may be implicated but their functional role remains to be experimentally addressed.

---

## [Author Response]

[…] This is a very nice manuscript reporting clear and well-controlled experiments that reveal an important function of TRAF2 in the maintenance of skin homeostasis. Also, the data on the (lack of) function of SPHK1 in TNF signalling is very convincing. The authors should discuss more carefully the skin pathology developing in their mice comparing with human psoriasis in terms of histological findings, the immune infiltrates, gene expression as well as the genetic studies.

Reviewer #1 (Minor Comments):

1) The Materials and methods is missing information about how TUBE experiment was conducted, and source of RIPK1 and cIAP antibodies.

Thanks for picking this up, we have now included the full information for the method and the source of the antibodies (subsection “Immuno precipitation of ubiquitin-conjugates (TUBE IP)”).

2) Figure 2 is missing blot for levels of anti ERK, JNK and p38 (A) and anti JNK (E).

We repeated this experiment using the same time points as before and tested with a full panel of antibodies and the appropriate loading controls (new Figure 2 and Figure 12). As before, IκBα degradation is complete in *Sphk1*^-/-^ and wild type MEFs but incomplete in *Traf2^-/-^*MEFs (n=1 primary MEF, n=4; comprising 3 biologically independent transformed MEF lines, one of which was tested twice). Likewise, JNK phosphorylation is maximal in wild type and *Sphk1*^-/-^ MEFs and is reduced and delayed in *Traf2^-/-^*MEFs (n=3). In the previous figure we had included phospho-p38 and phospho-ERK blots, however, in repeating experiments with different sources of MEFs for the revision, we observed variations in these responses. Therefore we cannot draw conclusions based on this data and have removed them. This fits with our poor experience with different biological MEF lines and is the reason that we prefer to use the far more reliable dermal fibroblasts (e.g. Figure 3).

Author response image 1.Primary (Top left) and transformed MEFs generated from the indicated knock-out mice stimulated with TNF for the indicated times and Western blotted with the indicated antibodies.These indicate the reproducibility of the results that we have shown in our final manuscript.**DOI:**
http://dx.doi.org/10.7554/eLife.10592.016

3) The authors should be more consistent in all blots regarding time points when they compare effects of WT, Traf2^-/-^ and Sphk1^-/-^ cells.

In Figure 2 we analyse 15, 60 and 120 minutes following TNF stimulation, which are well established time points that allow us to monitor activation and the return to baseline. In Figure 2 we also include time points of 4 and 6 hr because keratinocytes are a particularly important cell type in the study and we thought it relevant to investigate a prolonged response. In Figure 3, using the more involved TUBE protocol in MDFs we examined the early time

point of 5 minutes because ubiquitylation occurs as an early during the signalling cascade. When we saw a defect in ubiquitylation in *Traf2^-/-^*cells we extended the analysis to 60 minutes (old Figure 3) because that is where we started to observe NF-κB activation in *Traf2^-/-^*MDFs (Figure 2). Because we saw a

delayed RIPK1 ubiquitylation in *Traf2^-/-^*cells (old Figure 3) this led us to question whether *Sphk1*^-/-^ cells might also activate ubiquitylation aberrantly and therefore we performed a more detailed analysis of 15, 30, 60, 90 and 120 minutes in these cells (old Figure 3).

In response to the reviewer's comment we have now analysed the 0, 15, 60 and 120 minute time points following TNF stimulation of *Traf2^-/-^*and *Sphk1*^-/-^ MDFs and compared to wild type MDFs in the same experiment (New Figure 3). The reviewers will appreciate that these new results are identical with the old results and entirely consistent with the other data that we show in Figure 2κBα degradation, caspase-8 activation and ERK phosphorylation). Therefore we have removed the old Figure 3 and replaced these panels with our new Figure 3.

4) In Figure 3, why is there no observed caspase-8 ubiquitination in WT and Sphk1^-/-^ cells upon TNF stimulation, since cIAPs and TRAF2 (Gonzalvez et al., 2012) should be active? The authors may consider discussing this issue.

In Gonzalvez et althe authors use TRAIL or the Fas activating antibody Jo2 in HCT116 *Bax*^-/-^ cells to examine ubiquitylation of caspase-8, which they observed most strongly 1 hr post stimulation. In Figure 3 we analysed 5 minutes post TNF stimulation in MDFs. In our new experiment (new Figure 3), we extended our analysis to 120 minutes and also do not see signs of an ubiquitylated form of caspase-8 in wild type and *Sphk1*^-/-^ cells. However we think the completely different treatments and cell types used in these two works preclude us from drawing any conclusion or adding anything productive to the Discussion.

5) Figure 3: the authors should include also TUBE IP experiment with longer time points for Traf2^-/-^ which would confirm their hypothesis about delayed NF-kB activation.

As requested we have repeated this experiment (new Figure 3), which repeats and extends our previous observation that ubiquitylation of RIPK1 and phosphorylation of ERK is delayed and weaker in *Traf2^-/-^*MDFs compared to wild type or *Sphk1*^-/-^ cells.

6) Figure 6: the authors should add FACS contour plot and corresponding bar diagram for CD11b+Ly6C+ infiltrating neutrophils in Traf2^EKO^Tnf^-/-^ as they did for other cell suspensions, and compare the data with results for CD4^+^ T cells.

We have added the requested data quantifying CD11b Ly6G neutrophils from the epidermis of *Traf2^EKO^Tnf*^-/-^ mice (new panel in Figure 6 and new data in 6B). We have also now added flow cytometry plots to examine and quantify recruitment of neutrophils into the dermis of *Traf2^EKO^*and *Traf2^EKO^Tnf*^-/-^ mice compared with wild type dermis (New Figure 6).

7) The authors should focus in Discussion on experiments with IL-17 and discuss potential other signals for activation of non-canonical NF-kB independent of TNF.

In the Discussion section, we have added discussion on the clinical results with secukinumab and ixekizumab and the implications and limitations of our in vitroexperiments.

"IL-17 plays an important role in inflammatory diseases and monoclonal antibodies targeting IL-17 such as secukinumab and ixekizumab are performing well in psoriasis patients (Tse, 2013). […] However, in our experiments IL-17 did not increase *Traf2*^EKO/EKO^*Tnf*^-/-^ keratinocyte proliferation compared to *Traf2*^EKO/+^*Tnf*^-/-^ cells in vitroand loss of TRAF2 had no impact on IL-17 mediated activation of NF-κB and p38 in keratinocytes. While this does not exclude a role for IL-17 in the psoriasis observed in *Traf2^EKO^*or *Traf2^EKO^Tnf*^-/-^ mice we hypothesized that *Traf2EKOTnf*^-/-^ keratinocytes produced other factors to recruit inflammatory cells to the skin."

Likewise we have included a discussion on the implications of our results regarding non-canonical NF-κB signalling and psoriasis:

"This observation might be relevant to human psoriasis because it has been reported that the levels of TWEAK and its receptor Fn-14 are elevated in human psoriatic lesions (Cheng et al., 2015) and it is well established that TWEAK/Fn14 signalling induces degradation of TRAF2 and activation of non-canonical NF-κB signalling (Saitoh et al., 2003; Vince et al., 2008). Thus, our results suggest that inhibition of non-canonical NF-κB signalling might be beneficial in chronic inflammatory disease together with TNF inhibitions."

8) Figure 10: Kaplan-Meier graph does not depict final result with a survival of Traf2^EKO^Tnf^-/-^Nfkb2^-/-^ mice for up to 36 weeks. Instead, it is showing only survival percentage for other 6 genotypes. Since this survival curve is one of the most crucial results, the authors should add it.

Our apologies. The data was there but it was incorrectly colour coded (pink instead of blue) and was in an unfortunate position so that it gave the impression of demarking the graph boundary. We have now changed the format of this graph so that it is clear which survival line corresponds to which mouse strain. We have also substantially increased the numbers of the strains that are most directly relevant, namely *Traf2^-/-^Tnf*^-/-^*Nfkb2*^-/-^ (from 4 to 13) and *Traf2^-/-^Tnf*^-/-^*Nfkb2*^-/+^ (from 9 to 15). These new data also incorporate three *Traf2^-/-^Tnf*^-/-^*Nfkb2*^-/-^ mice that die of other causes than psoriasis. These causes are most likely due to their immune deficiency and also occur in *Tnf*^-/-^*Nfkb2*^-/-^ and *Traf2^EKO/EKO^Nik(Map3K14)^aly/aly^*mice (Figure 9 and Figure 10).

9) The authors describe that loss of Traf2 mostly induced lesions in anterior parts of the body, while animals lacking active TRAF2 and functional non-canonical NF-kB signaling were more susceptible to develop lesions in ventral and posterior parts. They should discuss possible reasons for these differences.

We note that we also see the same lesions developing in *Traf2^EKO/EKO^Nfkb2*^-/-^*Tnf*^-/-^ and *Tnf*^-/-^*Nfkb2*^-/-^ mice (Figure 10). NIK and non-canonical NF-κB signalling play an important role in immune responses and in establishing the immune system. Given the location of the lesions in *Traf2^EKO/EKO^Nfkb2*^-/-^ and *Traf2^EKO/EKO^Nfkb2*^-/-^*Tnf*^-/-^ mice, we think it likely that these lesions are connected with a failure to correctly regulate responses to bacteria in these locations. This is consistent with the other report that fails to generate complete *Traf2*^-/-^*Map3k14*^-/-^ or *Traf2*^-/-^*Map3k14*^-/-^*Tnf*^-/-^ mice (Lin, PNAS 2011) that we have cited.

10) In the last Results paragraph, the authors twice talk about the phenotype of Traf2^EKO^/+Tnf^-/-^Nfkb2^-/-^ which isn't presented nowhere on figures. Please, provide figures showing these results or correct that paragraph.

These mice were shown in Figure 11, which has now been merged with Figure 10. We have provided the appropriate figure reference (Figure 10) for the statement concerning these mice in the text. One of the references to this strain of mice, in the last sentence of the Results section, was however a typo. It should have referred to *Traf2^EKO/EKO^Tnf^-/-^Nfkb2^-/-^*and we have now fixed this error.

Reviewer #2 (Minor Comments):

In Figure 1 and Figure 2, the β actin blots are not equalized among the lanes. It probably will not change the conclusions, but loading the same amount of samples to each lane will make the figures look more convincing.

In Figure 1 the actin signal was burnt out due to over-exposure. We have replaced this loading control with a shorter exposure time.

We repeated the experiment in Figure 2 and the new experiment has loading controls for JNK and a more representative actin blot.

Reviewer #3 (Minor Comments):

The authors state that the triple mutant mice (TRAF2EKO, Tnf^-/-^, Nfkb2^-/-^) 'occasionally' developed skin lesions but do not indicate how many mice have been analysed and how many developed lesions. Please clearly state the number of mice analysed for each genotype. It would also be helpful to clearly state the age of mice shown in all figures.

We have discussed and included pictures of these mice and provided the pertinent information about the age of the mice:

"Three out of 13 of the *Traf2^EKO^Tnf^-/-^Nfkb2^-/-^*mice developed skin lesions at an early age around the mouth and anus (Figure 10). This is likely to be due to the susceptibility of NFκB2 deficient mice to opportunistic infections because both *Tnf^-/-^Nfkb2^-/-^*and *Traf2^EKO/EKO^Nfkb2^-/-^*mice succumbed to such infections (Figure 10 and Shinkura et al., 1996; Yin et al., 2001)."

There is a wrong citation in the Discussion "Several mouse models of skin inflammation have been rescued by depletion of TNF (Gerlach et al., 2011; Nenci et al., 2006; Pasparakis, Schmidt-Supprian, and Rajewsky, 2002)." The citation Pasparakis, Schmidt-Supprian, and Rajewsky, 2002 is not the right one, the correct one is Pasparakis, Courtois et al., 2002.

Thank you for this correction.

Figure 6 in the subsection “Psoriasis-like inflammation in Traf2^EKO^ mice is characterised by skin infiltration of neutrophils and IFNγ producing CD4^+^ T cells in” referring to the data on IFNγ+ T cells should be Figure 6.

The whole figure has changed and we have endeavoured to cite the new figure correctly.

[Editors' note: further revisions were requested prior to acceptance, as described below.]

The inflammatory skin lesions developing in mice with keratinocyte-specific TRAF2 knockout can be described as psoriasis-like, but it is not correct to write that these mice develop psoriasis, as in the following instances in the manuscript:

In the subsection “Psoriasis-like inflammation in Traf2^EKO^ mice is characterised by skin infiltration of neutrophils and IFNγ producing CD4^+^ T cells in”: "However, both wild type and Ifng^-/-^ chimeras developed skin inflammation to the same extent indicating that IFNγ is not a major contributor to the psoriasis (Figure 7)."

In the subsection “Blocking non-canonical NF-κB signalling and depletion of TNF together prevent inflammation caused by TRAF2 deficiency in keratinocytes”: "Contrary to our hypothesis, deficiency in Map3k14 or Nfkb2 did not prevent the psoriasis in Traf2^EKO^ mice (Figure 9 and Figure 10)."

Discussion: "Unfortunately, we were unable to age these mice more than 15 weeks as they developed lymphadenopathy and could therefore not examine the effect of Mlkl^-/-^Casp8^-/-^ on the late onset psoriasis."Discussion: "While this does not exclude a role for IL-17 in the psoriasis observed in Traf2^EKO^ or Traf2^EKO^Tnf^-/-^ mice we hypothesized that Traf2^EKO^Tnf^-/-^ keratinocytes produced other factors to recruit inflammatory cells to the skin."

Thanks for pointing out this. We have changed all the above items to the appropriate and relevant terms to the phenotype of *Traf2^EKO^*mice, which are consistent with the title and rest of the manuscript.

*In Figure 11, the schematic model gives the impression that in the absence of TNF the inflammatory skin lesions are induced by M-CSF and IL-23 released by keratinocytes and IFNγ produced by T cells. As there is no data in the paper supporting causal function of these cytokines, it would be more appropriate not to refer to specific cytokines or to at least clearly describe in the scheme and the legend that these cytokines may be implicated but their functional role remains to be experimentally addressed.*

Now we have changed and improved Figure 11 and its legend addressing the mentioned issues. We have changed the tile of the figure to the “Proposed mechanisms by which loss of TRAF2 in keratinocytes causes a psoriasis-like skin inflammation.” We have changed the specific cytokines to a general term of “inflammatory cytokines”.